# A space hurricane over the Earth's polar ionosphere

Qing-He Zhang [1✉], Yong-Liang Zhang[2], Chi Wang[3], Kjellmar Oksavik [4,5], Larry R. Lyons[6], Michael Lockwood [7], Hui-Gen Yang[8], Bin-Bin Tang [3], Jøran Idar Moen[9,5], Zan-Yang Xing[1], Yu-Zhang Ma[1], Xiang-Yu Wang[1], Ya-Fei Ning[10] & Li-Dong Xia[1]

In Earth's low atmosphere, hurricanes are destructive due to their great size, strong spiral winds with shears, and intense rain/precipitation. However, disturbances resembling hurricanes have not been detected in Earth's upper atmosphere. Here, we report a long-lasting space hurricane in the polar ionosphere and magnetosphere during low solar and otherwise low geomagnetic activity. This hurricane shows strong circular horizontal plasma flow with shears, a nearly zero-flow center, and a coincident cyclone-shaped aurora caused by strong electron precipitation associated with intense upward magnetic field-aligned currents. Near the center, precipitating electrons were substantially accelerated to ~10 keV. The hurricane imparted large energy and momentum deposition into the ionosphere despite otherwise extremely quiet conditions. The observations and simulations reveal that the space hurricane is generated by steady high-latitude lobe magnetic reconnection and current continuity during a several hour period of northward interplanetary magnetic field and very low solar wind density and speed.

[1] Shandong Provincial Key Laboratory of Optical Astronomy and Solar-Terrestrial Environment, Institute of Space Sciences, Shandong University, Weihai, Shandong, PR China. [2] The Johns Hopkins University Applied Physics Laboratory, Laurel, MD, USA. [3] State Key Laboratory of Space Weather, Center for Space Science and Applied Research, Chinese Academy of Sciences, Beijing, PR China. [4] Birkeland Centre for Space Science, Department of Physics and Technology, University of Bergen, Bergen, Norway. [5] The University Centre in Svalbard, Longyearbyen, Norway. [6] Department of Atmospheric and Oceanic Sciences, University of California, Los Angeles, CA, USA. [7] Department of Meteorology, University of Reading, Reading, UK. [8] Ministry of Natural Resources Key Laboratory of Polar Science, Polar Research Institute of China, Shanghai, PR China. [9] Department of Physics, University of Oslo, Blindern, Oslo, Norway. [10] School of Microelectronic, Shandong University, Jinan, Shandong, PR China. ✉email: zhangqinghe@sdu.edu.cn

Hurricanes often cause loss of life and property through high winds and flooding resulting from the coastal storm surge of the ocean and the torrential rains[1,2]. They are characterized by a low-pressure center (hurricane eye), strong winds and flow shears, and a spiral arrangement of towering clouds with heavy rains[1,3]. In space, astronomers have spotted hurricanes on Mars, and Saturn, and Jupiter[4,5], which are similar to terrestrial hurricanes in the low atmosphere. There are also solar gases swirling in monstrous formations deep within the sun's atmosphere, called solar tornadoes with widths of several Earth radii ($R_E$)[6]. However, hurricanes have not been reported in the upper atmosphere of the planets in our heliosphere. Although vortex structures of aurora, called auroral spirals, often appear to evolve from arc-like auroras to a train of two or more spirals of diameter ~50 km in the Earth's nightside auroral oval (about 65–75° magnetic latitude)[7,8], they are not unusually intense and do not have similar features of a typical hurricane. In the Earth's polar cap region (about 75–90° magnetic latitude), high-latitude dayside auroral (HiLDA) spots, but without spiral or hurricane features, have been reported to be caused by precipitating electrons predominantly during northward interplanetary magnetic field (IMF) with a strongly positive IMF $By$ component[9–13].

A hurricane is clearly associated with strong energy and mass transportation, so a hurricane in Earth's upper atmosphere must be violent and efficiently transfer solar wind/magnetosphere energy and momentum into the Earth's ionosphere. It is well known that magnetic reconnection and Kelvin–Helmholtz (K–H) instability are the most important and fundamental processes for coupling solar wind energy into the magnetosphere-ionosphere system and similar coupling occurs in other astrophysical, space, and laboratory plasmas. For a southward IMF (which occurs nearly half of the time), magnetic reconnection occurs at the low-latitude dayside magnetopause[14–16] and it directly brings solar wind energy and plasma into the magnetosphere[17–20]. Under a northward IMF condition, magnetic reconnection is limited to a small high latitude region and K–H instability becomes important in bringing solar wind energy and plasma into the magnetosphere when the solar wind density and speed are high[21–27]. It is generally believed that transfer of solar wind energy and plasma into the magnetosphere and ionosphere is very weak when geomagnetic activity is extremely quiet (such as during a long period of strongly northward IMF with very low solar wind density and speed).

Here, we present an observation of a long-lasting, large and energetic space hurricane in the northern polar ionosphere that deposited solar wind/ magnetosphere energy and momentum into the ionosphere during a several hour period of northward IMF and very low solar wind density and speed.

## Results

**Interplanetary and geomagnetic conditions**. On 20th August 2014, a relatively stable northward IMF condition (IMF $Bz > 0$ for more than 8 h) occurred with a large duskward component (IMF $By$ ~13 nT), and roughly stable interplanetary conditions with low solar wind speed and density (Fig. 1a–c). The IMF $Bx$ and $Bz$ decreased slowly from 10 to 5 nT over the 8-h period, and the low solar wind speed (around 340 km/s) and number density (around 2 cm$^{-3}$) indicates a very low dynamic pressure of around 0.5 nPa (gray shading in Fig. 1 indicates the interval of interest). These conditions are not favorable for magnetic reconnection at the low-latitude dayside magnetopause[14–16], nor for triggering of the K–H instability between the solar wind and magnetosphere in the magnetospheric flank regions[21–25], but are suitable for forming high-latitude dayside auroral spots in the polar cap region[9–13]. The symmetric ring current H index (SYM-H) and

auroral electrojet AL and AU indices show non-storm and quiet auroral oval geomagnetic activity during the interval of interest (Fig. 1d, e).

**Observations of the space hurricane**. Figure 2a shows an example of auroral observations from the Defense Meteorological Satellite Program (DMSP)[28] F16 Special Sensor Ultraviolet Spectrographic Imager (SSUSI) over the Northern Hemisphere. Around the north magnetic pole, a cyclone-like auroral spot (diameter over 1000 km) with multiple arms and a trend of anti-clockwise rotation is analogically named as space hurricane hereafter (Supplementary Movie 1). The space hurricane was observed by four DMSP satellites, and the observed flows at all the spacecraft magnetic local times (MLTs) were consistent with circular fast flows surrounding the hurricane center (Supplementary Movie 1). It appeared in the polar cap after multiple transpolar arcs disappeared when the interplanetary conditions changed from strongly northward dominated IMF ($Bz = $~17 nT, $By < 5$ nT) with comparable solar wind number density ($Nsw = $~4 cm) to the conditions described above (see Fig. 1 and Supplementary Movie 1), similar with the conditions for the appearance of the HiLDA spots[9,13]. There is no conjugate auroral spot in the Southern Hemisphere (Supplementary Movie 2), as expected from the direction of circulation of plasma within the polar cap ionosphere under strong IMF $By$ conditions[9,12,13,29,30]. The space hurricane lasted about 8 h, gradually decayed and merged into the duskside auroral oval around 20:00 UT when the IMF turned southward (see Fig. 1 and Supplementary Movie 1 and 2), same as the disappearance of the HiLDA spots[9]. In addition, the auroral oval (between 70° and 85° MLAT) was generally quiet in the dawn sector while strong arcs persisted in the dusk sector. The field-aligned current (FAC) along the satellite track calculated from the magnetic field measurements of DMSP F16 indicates that the space hurricane was associated with an upward FAC.

Around the space hurricane, the Active Magnetosphere and Planetary Electrodynamics Response Experiment (AMPERE) global FAC map (Fig. 2b), estimated from magnetic main-field perturbations observed by Iridium engineering magnetometers[31,32], also shows a spot-like strong upward FAC (red, reaching above 1.5 μAm$^{-2}$) within a negative electric potential cell (contours in Fig. 2b), which is co-located with the space hurricane and confirms that the space hurricane is surrounded by circular convection flow. This circularity or vorticity of the flow includes the flow shears and the flow curvature. The flow shear is approximately constant, but the curvature increases towards the hurricane center, thus forming the spot-like FAC. The FAC spot was surrounded and closed by downward cusp FAC on its equatorward side (blue, reaching about −1.5 μAm$^{-2}$, Fig. 2b), so that the combination of hurricane and cusp currents maintained current continuity in the ionosphere[33]. The FAC spot also lasted for more than 8 h (with sometimes a FAC hole developed in the center, see Supplementary Movie 1), and merged into the classical Region 1 FAC about 20:00 UT when the IMF turned southward (see Supplementary Movie 1). Note that upward FACs also appeared to be associated with the duskside auroral arcs, but they are much weaker than the FAC spot.

The drift vectors (perpendicular to the spacecraft orbit) in Fig. 2a (mauve) and Fig. 3a show the cross-track horizontal (nearly north-south direction) ionospheric plasma drift from DMSP F16. These show that the space hurricane had zero horizontal flow near its center (the hurricane eye) as well as strong flow shears around the edges: strong sunward flows on its duskside (maximum ~2100 m/s) and antisunward flows on its dawnside (maximum ~800 m/s). Note that there will be a small horizontal offset between the in situ plasma drift data and the

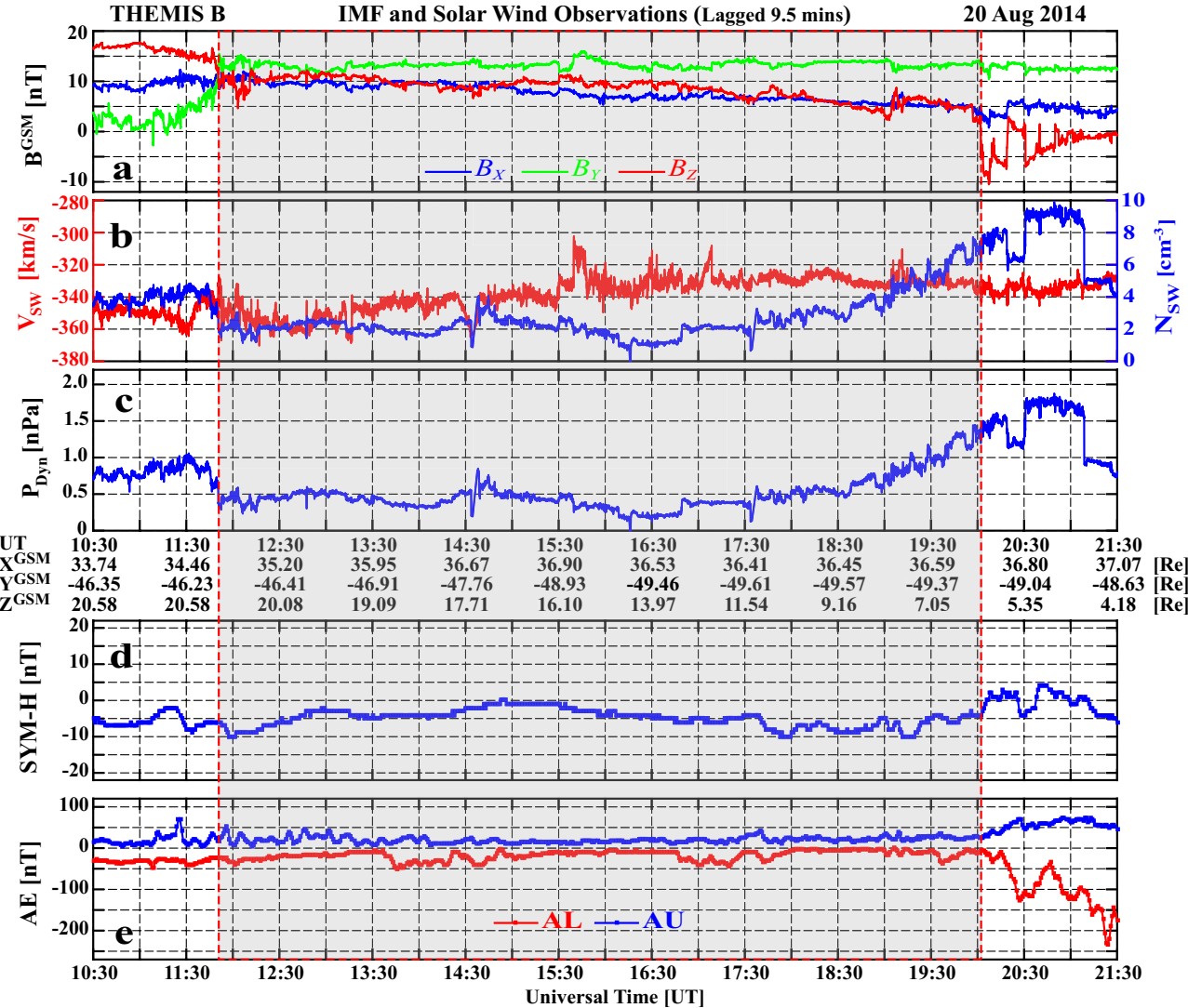

**Fig. 1 An overview of the interplanetary conditions and geomagnetic indices on 20th August 2014. a** The IMF components in geocentric solar magnetosphere (GSM) coordinates; **b** the solar wind number density and speed; **c** the solar wind dynamic pressure, $P_{Dyn}$; **d** the provisional SYM-H geomagnetic index (from 6 stations); and **e** the provisional auroral electrojet geomagnetic indices (from 11 stations): red and blue lines are for AU and AL. Interplanetary data is measured by the Time History of Events and Macroscale Interactions during Substorms (THEMIS)[44] B satellite (in the moon orbit), and has been lagged by 9.5 min to the dayside magnetopause.

auroral images, because the converging magnetic field will cause the flow shears to decrease in horizontal extent from the DMSP in situ observation altitude (860 km) to the auroral mapping altitude (110 km, Fig. 2a). These flow shears give a clockwise circulation of ionospheric flow, which appears opposite to the rotation trend that might be inferred from the multiple arms of the auroral spot. This indicates an interesting difference from tropospheric hurricanes that is discussed latter.

Figure 3b–e shows that the space hurricane is also associated with ion upflows, enhanced electron temperature (about 1000 K enhancement), a negative-to-positive bipolar magnetic structure (implying a circular magnetic field perturbation) and strong upward field-aligned currents (consistent with the AMPERE FAC observations). Within the space hurricane, the total energy flux (JE) and the average energy of the precipitating electrons were significantly increased (Fig. 3f, g), resulting in a time integrated JE ($\Sigma$JE) up to $2.48 \times 10^{14}$ eV/(cm$^2$·sr) from 16:16:58 to 16:18:51 UT, which is about 91.49% of the $\Sigma$JE ($2.71 \times 10^{14}$ eV/(cm$^2$·sr)) for the whole polar pass (see Tables 1 and 2). The $\Sigma$JE ($2.71 \times 10^{14}$ eV/(cm$^2$·sr)) is about 10 times higher than that for a polar pass

without a space hurricane also under a geomagnetic quiet condition (see Tables 1 and 2 for the DMSP pass from 08:54:22 to 09:11:24 UT on 21 June 2010). It is about 4.6 times higher than that for a pass under typical southward IMF conditions during non-storm time (see Tables 1 and 2 for the DMSP pass from 16:26:00 to 16:46:00 UT on 08 October 2014). Furthermore, it is only about 4.6 times smaller than the $\Sigma$JE of a pass during the main phase of the first super geomagnetic storm of solar cycle 24 which had intense solar wind driving and strong southward IMF (see Tables 1 and 2 for the pass from 23:14:00 to 23:44:00 UT on 17 March 2015). The space hurricane has an average energy flux about 5.5 times higher than its own whole polar pass, and this whole pass is about 15.1 times higher than the pass for the typical quiet case, about 8.3 times higher than the pass for the typical southward IMF case, and even 3.2 times higher than the super storm case (see Table 2). These means that the average electron energy flux in the space hurricane ($2.2 \times 10^{12}$ eV/(cm$^2$·s·sr)) is much higher than that during substorm expansion[34], but is comparable to that during super storms (sometimes exceeding $10^{13}$ eV/(cm$^2$·s·sr) during strikingly super storms)[35,36]. Note that

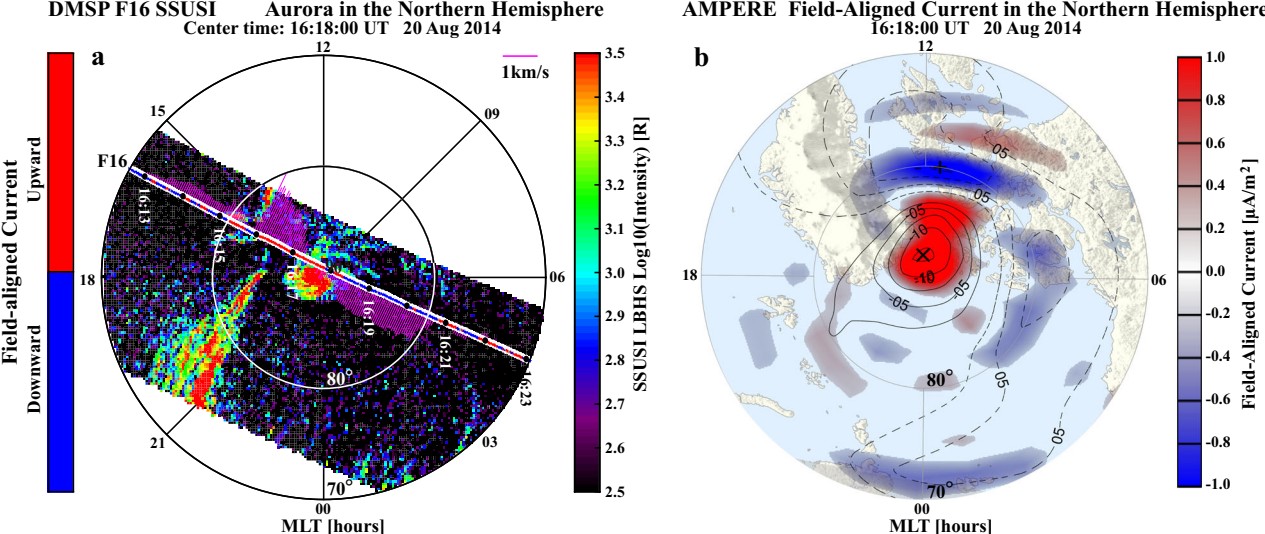

**Fig. 2 An example of aurora and FACs observations in the polar region of the Northern Hemisphere. a** Aurora in the Lyman–Birge–Hopfield short-band (LBHS) band (wavelength of 140–150 nm), the measured cross-track horizontal ion flows shown in mauve drift vectors perpendicular to the orbit, and the sign of the FACs shown in red and blue color along the satellite track. The aurora is observed by the SSUSI instrument on board the DMSP F16 satellite, the ion flow is measured by the special sensor for ions, electrons, and scintillation (SSIES) and the FAC is calculated from the magnetic field measurement of special sensor microwave (SSM) instrument. These instruments are all on board the DMSP F16 satellite. **b** The distribution map of the FACs and potential of AMPERE magnetic perturbation data products derived from the Iridium satellites constellation.

the large electron precipitation flux during substorms and storms is within the auroral oval, which is located at much lower latitudes than the space hurricane. Table 2 also shows that the average energy flux in the space hurricane ($2.2 \times 10^{12}$ eV/(cm²·s·sr)) is about 28.8 times higher than that in the auroral oval ($7.8 \times 10^{10}$ eV/(cm²·s·sr)) and 71.7 times higher than that in the diffuse aurora region ($3.1 \times 10^{10}$ eV/(cm²·s·sr)) during the same pass (see Fig. 3g). The space hurricane also has the highest maximum and average energy in the magnetic pole region compared to the values during typical quiet and super storm times in the same region (see Table 2). These indicate that the space hurricane leads to large and rapid deposition of energy and flux into the polar ionosphere during an otherwise extremely quiet geomagnetic condition, suggesting that current geomagnetic activity indicators do not properly represent the dramatic activity within space hurricanes, which are located further poleward than geomagnetic index observatories.

Clear electron inverted-V acceleration appeared within the space hurricane with ~10 keV energy electron precipitation near the hurricane center and ~1 keV energy electron precipitation around the edge (Fig. 3g, h), the amount of electron energization increasing with increasing upward FAC strength due to an increasing field-aligned potential drop. Under this quasi-steady condition with uniform ionospheric conductivity due to sunlit conditions, the large-scale, stronger FACs near the hurricane center should be connected to convergent ionospheric Pedersen currents caused by the combination of the velocity shear and the curvature of the circular flow increasing towards the hurricane center, inferring that a FAC spot or funnel with circular fast flows appears in the electron source region. Note that there is almost no ion precipitation in the space hurricane area (Fig. 3i) and no conjugate auroral structure in the Southern Hemisphere (see Supplementary Movie 2), same as for HiLDA spots[9,12]. These observations indicate that the space hurricane contains accelerated electron precipitation that likely originated from the open-magnetic field, high-latitude lobe region of the magnetosphere.

The observed features and formation conditions of the space hurricane are almost the same as for the HiLDA spots from

coincident observations by the IMAGE and FAST satellites[9–13]. This indicates that HiLDA spot may be the same phenomenon as the space hurricane in the polar cap region. However, the important characteristics of the space hurricane identified here, i.e., a cyclone-shaped aurora, a strong circular horizontal plasma flow with shears, and a nearly zero-flow center, could not be identified in the previous HiLDA observations[9–13] due to the relatively low spatial resolution in that auroral image data and the lack of coincident ionospheric plasma drift measurements.

**Data-driven simulation.** The formation of space hurricane is further investigated by simulation using a high-resolution 3-D global magnetohydrodynamics (MHD) code, piecewise parabolic method[37] with a Lagrangian remap to MHD (PPMLR-MHD)[38,39], which uses the measured interplanetary conditions as inputs. Figure 4a shows a 3-D view of simulated FACs in the GSM X–Z plane and X–Y plane at Z = 8 $R_E$. The Sun is on the right. The magnetopause boundary is characterized by a narrow downward FAC belt (purple) on the dayside, and by a narrow upward FAC belt (red) on the dawn flank and in the high-latitude lobe region. In the center of Fig. 4a, a strong upward FAC funnel appears to nearly link the polar ionosphere to the inner edge of the high-latitude magnetopause FAC belt. The 3D topology of selected magnetic field lines suggests that there is magnetic reconnection occurring between the IMF and Earth's magnetic field at the dayside magnetopause around both the tailward (red lines) and equatorward (light blue lines) field lines of the cusp (Fig. 4b). The reconnected open field lines link to the northern hemisphere, and tend to move dawnward and then tailward from the morning side to the afternoon side in the high-latitude lobe region (highlighted by the colored and numbered field lines and an arrowed curve in Fig. 4b). Figure 4c shows the upward FAC closing through a strong downward FAC band on the dawn side that appears to connect to the downward FAC belt of the dayside magnetopause. These are remarkably consistent with the AMPERE and DMSP FAC observations. The funnel of FAC appears as a spot with several arms and a trend of anti-clockwise rotation (Fig. 4d,

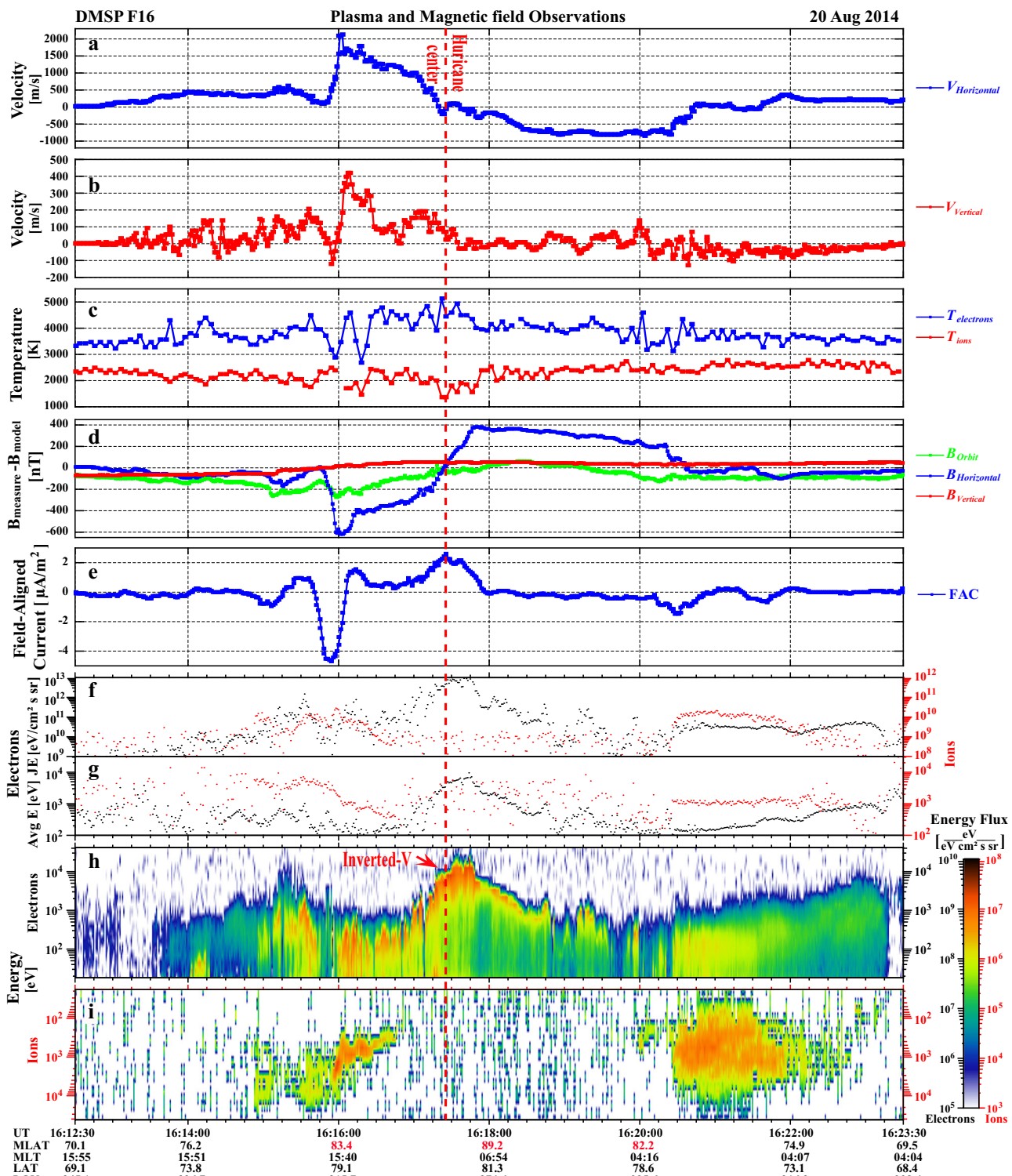

**Fig. 3 The in-situ plasma and current conditions for the orbit of DMSP F16 shown in Fig. 2a. a, b** The cross-track horizontal and vertical ion flow; **c** electron and ion temperature; **d** the three components of the measured magnetic field subtracted by the modeled magnetic field from the International Geomagnetic Reference Field (IGRF) model[45]; **e** the calculated field-aligned current; **f** the precipitating electron and ion total energy flux, JE; **g** the electron and ion average energy, Avg E; **h** the precipitating electron energy flux, and **i** the precipitating ion energy flux. Data in **a–c** are measured by SSIES, data in **d**, **e** are observed or calculated from the magnetic field measurement of SSM, and data in **f–i** are measured by the Special Sensor for Precipitating Particles (SSJ4) instrument on board the DMSP F16 satellite.

Supplementary Movie 3 and Supplementary Fig. 1), consistent with the DMSP SSUSI auroral and plasma observations. These upward FACs (both from the simulation and observations) cause magnetic field-aligned acceleration of magnetospheric electrons

(probably through the Knight current–voltage process to keep current continuity[11,25,33,40,41]) that precipitate into the polar ionosphere and generate the hurricane structure in the aurora. Note that a pair of current sheets can be seen on the duskside of

**Table 1 The average values of the interplanetary and geomagnetic conditions for four typical conditions.**

| Typical Conditions | Time intervals | IMF Bz [nT] | Nsw [cm$^{-3}$] | Vsw [km/s] | Dst [nT] | AL [nT] |
|---|---|---|---|---|---|---|
| Extremely quiet time with space hurricane | 16:12:30–16:23:30 20 Aug 2014 | ~8 | ~2 | ~340 | ~ −10 | ~ −20 |
| Typical Quiet time without space hurricane | 08:54:22−09:11:24 21 June 2010 | ~5 | ~3 | ~390 | ~ −3 | ~ −25 |
| Typical southward IMF case for non-storm time | 16:26:00–16:46:00 08 October 2014 | ~ −7 | ~5 | ~352 | ~ −20 | ~ −495 |
| Super storm time | 23:14:00−23:44:00 17 March 2015 | ~ −18 | ~7 | ~549 | ~ −222 | ~ −1500 |

Column 1 is the typical conditions, Column 2 is the time intervals for the typical regions during the DMSP crossings, Column 3–7 are the average values of the IMF Bz, Nsw, Vsw, Dst, and AL indexes, respectively.

**Table 2 The energy flux and average energy of the precipitating electrons observed by SSJ4 instrument onboard the DMSP satellites under different conditions.**

| Typical regions | Time intervals | Duration [s] | ΣJE [eV/ (cm$^2$ sr)] | EnF$_{avg}$ [eV/ (cm$^2$ s sr)] | PΣJE | E$_{avg}$ [eV] | E$_{max}$ [eV] |
|---|---|---|---|---|---|---|---|
| Space hurricane | 16:16:58–16:18:48 | 110 | $2.48 \times 10^{14}$ | $2.25 \times 10^{12}$ | 91.49% | $2.27 \times 10^3$ | $9.56 \times 10^3$ |
| Auroral oval | Duskside: 16:14:52–16:16:49 Dawnside: 16:20:23–16:21:57 | 211 | $1.64 \times 10^{13}$ | $7.81 \times 10^{10}$ | 6.08% | 248 | $1.35 \times 10^3$ |
| Diffuse aurora | 16:21:57–16:23:16 | 79 | $2.51 \times 10^{12}$ | $3.14 \times 10^{10}$ | 0.93% | 616 | $2.64 \times 10^3$ |
| Whole North polar pass for extremely quiet time with space hurricane | 16:12:30–16:23:30 | 660 | $2.71 \times 10^{14}$ | $4.10 \times 10^{11}$ | 100% | 709 | $9.56 \times 10^3$ |
| Whole North polar pass for typical Quiet time without space hurricane | 08:54:22–09:11:24 21 June 2010 | 1022 | $2.79 \times 10^{13}$ | $2.72 \times 10^{10}$ | 100% | 847 | $1.58 \times 10^4$ |
| Whole North polar pass for Typical southward IMF case for non-storm time without space hurricane | 16:26:00–16:46:00 08 October 2014 | 1200 | $5.92 \times 10^{13}$ | $4.93 \times 10^{10}$ | 100% | 924 | $1.78 \times 10^4$ |
| Whole North polar pass for super storm time without space hurricane | 23:14:00–23:44:00 17 March 2015 | 1800 | $1.25 \times 10^{15}$ | $6.97 \times 10^{11}$ | 100% | 988 | $9.17 \times 10^3$ |

Column 1 is the typical regions, Column 2 is the time intervals for the typical regions during the DMSP crossings, Column 3 is the duration in seconds of the time intervals shown in Column 2, Column 4 is the time integrated total electron energy flux (ΣJE), Column 5 is the average electron energy flux (EnF$_{avg}$), Column 6 is the percentage of ΣJE (PΣJE), Column 7 is the electron average energy (E$_{avg}$), Column 8 is the maximum electron energy (E$_{max}$).

the spot, at ~18–21 MLT, which appear to correspond with the duskside auroral arcs seen in the DMSP SSUSI images. These consistencies provide strong evidence that the PPMLR-MHD model captures the key physical processes for these northward IMF conditions.

## Discussions

**Formation of the space hurricane.** Figure 5a schematically summarizes the main observational features of the space hurricane. A large cyclone-shaped auroral spot is shown with a nearly zero-flow center and strong circular horizontal plasma flow, shears, electron precipitation, and upward FACs. These features resemble a typical hurricane in the lower atmosphere. A circular large convection lobe-cell of the space hurricane as seen within the ionosphere is embedded within the normal afternoon convection cell, which is formed due to high-latitude lobe reconnections[9,12,14,26,29,30,42].

During a northward IMF with a dominant By component, magnetic reconnection occurs between IMF and the Earth's open-magnetic field lines tailward of the cusp in the afternoon sector[9,12,14,26,29,30] (high-latitude lobe reconnections, Fig. 4b and Supplementary Fig. 2 and Supplementary Movie 4). The newly reconnected open field lines are draped by the solar wind to move dawnward and then tailward from the morning side to the afternoon side in the high-latitude lobe region[26,29]. During their dawnward and tailward motion, an elongated FAC sheet forms due to the flow shear, and the magnetosheath ions precipitate into the cusp ionosphere along field lines to give the downward FACs (like traces of dropping sands from a moving hourglass). In order to maintain current continuity in the ionosphere, the system sets up an upward FAC with a parallel potential that accelerates the existing electrons into the ionosphere and creates an arm of the auroral spot[12,33] observed by DMSP F16 in Fig. 2a and shown in Fig. 5a.

When the lobe reconnection is pulsed or quasi-steady for an extended period of time (e.g., several hours), the reconnected open field lines will gradually return to their previous positions and participate in a new cycle of magnetic reconnection (Supplementary Movie 4), which will eventually form a cyclone-shaped funnel of FAC (see Fig. 5b) with multiple FAC arms and a clockwise circulation of the plasma flow, due to the pressure gradient and magnetic stresses on both sides of the funnel for completing the FACs and the flow shear and curvature of the circular flow. Inside the funnel, a corkscrew magnetic field forms with circular flow and upward FACs, which accelerate electrons that precipitate into the ionosphere[12,25,41] and create the auroral spot with multiple arms as observed by DMSP F16 in Fig. 2a. In other words, the auroral arms represent the trace of the footprints of the reconnected magnetic field lines, and shows an illusional trend of anti-clockwise rotation, which is opposite to the flow circulation and different from tropospheric hurricanes.

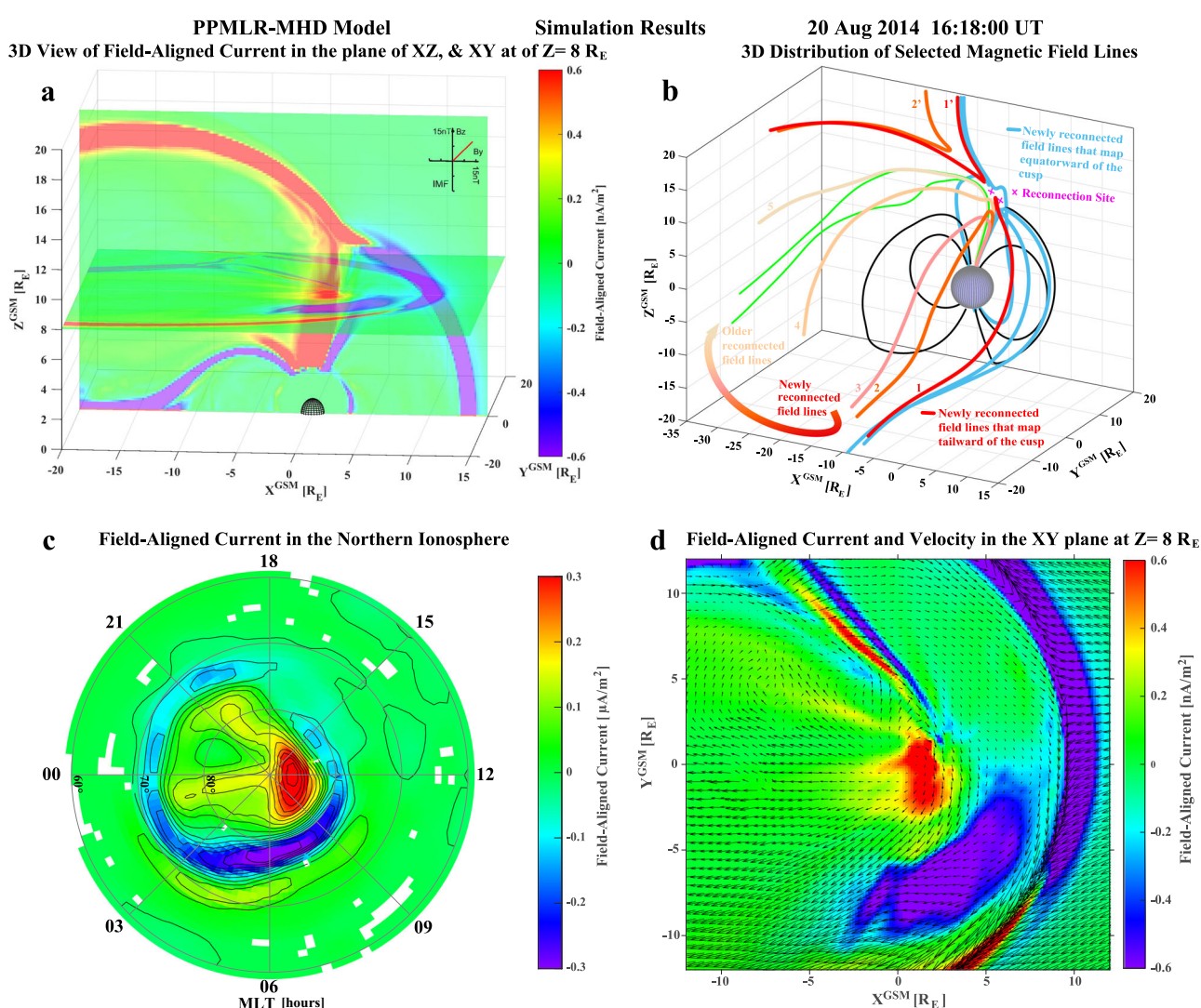

**Fig. 4 3-D and 2-D view of simulated FACs and selected magnetic field lines by the PPMLR-MHD code at the center time of the example in Fig. 2a. a** 3-D view of the simulated FACs in the GSM X–Z plane, and the X–Y plane at Z = 8 R$_E$; **b** 3-D distribution of selected magnetic field lines with magenta crosses representing the reconnection sites and the numbered field lines in red to light brown representing the newly to old evolution of the reconnected field lines that also highlighted by the thick arrowed color curve; **c** 2-D distribution of simulated FACs in the northern polar ionosphere with FAC contour lines, and **d** close-up view of the 2-D distribution of FACs and plasma velocity vectors in the X–Y plane at Z = 8 R$_E$.

This funnel becomes the most efficient channel to transfer the solar wind/magnetosphere energy and momentum into the ionosphere, and to accelerate terrestrial ions that escape into the magnetotail or interplanetary space, during periods of very low solar wind density and speed and a northward IMF with a dominant $By$ component. The footprint of these field line trajectories forms a circular large ionospheric convection lobe-cell with strong embedded circular horizontal plasma flow inside the normal afternoon convection cell[9,12,26,29,30,42]. Within this lobe-cell, strong radial electric fields point toward the cell center leading to a strong upward FAC that maintains current continuity in the ionosphere[25,33,41]. Strong magnetic field-aligned electric fields are required to give the strong FAC, accelerating electrons up to ~10 keV that precipitate and form the auroral signature of the space hurricane[25,41]. These observations indicate that there is a significant difference between the drivers of atmospheric and space hurricanes. Hurricanes or tropical cyclones require strong driving from below (latent heat flux due to rising moist air over a warm ocean), while space hurricanes occur under an extremely quiet interplanetary condition (low solar wind speed, density, and northward interplanetary magnetic field). The extremely quiet interplanetary condition results in efficient lobe reconnection which leads to the formation of the space hurricane. The space hurricane opens a rapid energy transfer channel from space to the ionosphere and thermosphere, and would be expected to lead to important space weather effects like increased satellite drag, disturbances in High Frequency (HF) radio communications, and increased errors in over-the-horizon radar location, satellite navigation, and communication systems[15,43]. The space hurricane is likely a universal phenomenon, occurring at other magnetized bodies in the universe (planets and their moons, etc.). The process may also be important for the interaction between interstellar winds and other solar systems throughout the universe.

## Methods
**PPMLR-MHD model**. The PPMLR-MHD model is a 3-D MHD model, which is based on an extension of the piecewise parabolic method[37] with a Lagrangian remap to MHD[38,39]. It is designed particularly for the solar wind–magnetosphere–ionosphere system[22–24]. The model possesses a high resolution for capturing MHD shocks and discontinuities and a low numerical dissipation for examining possible instabilities inherent in the system[22].

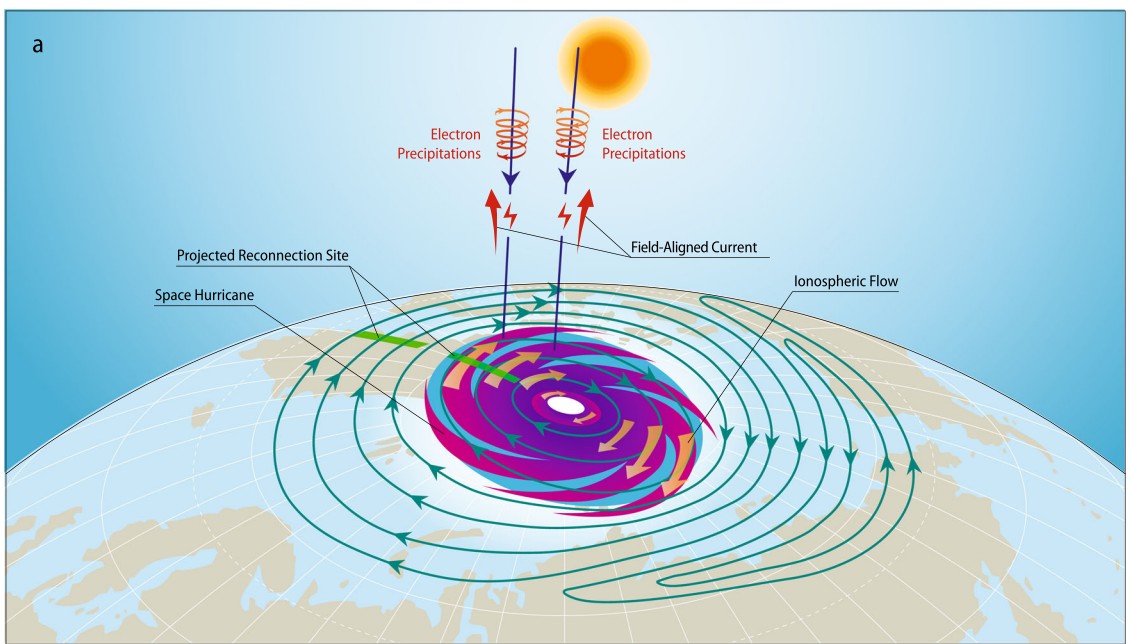

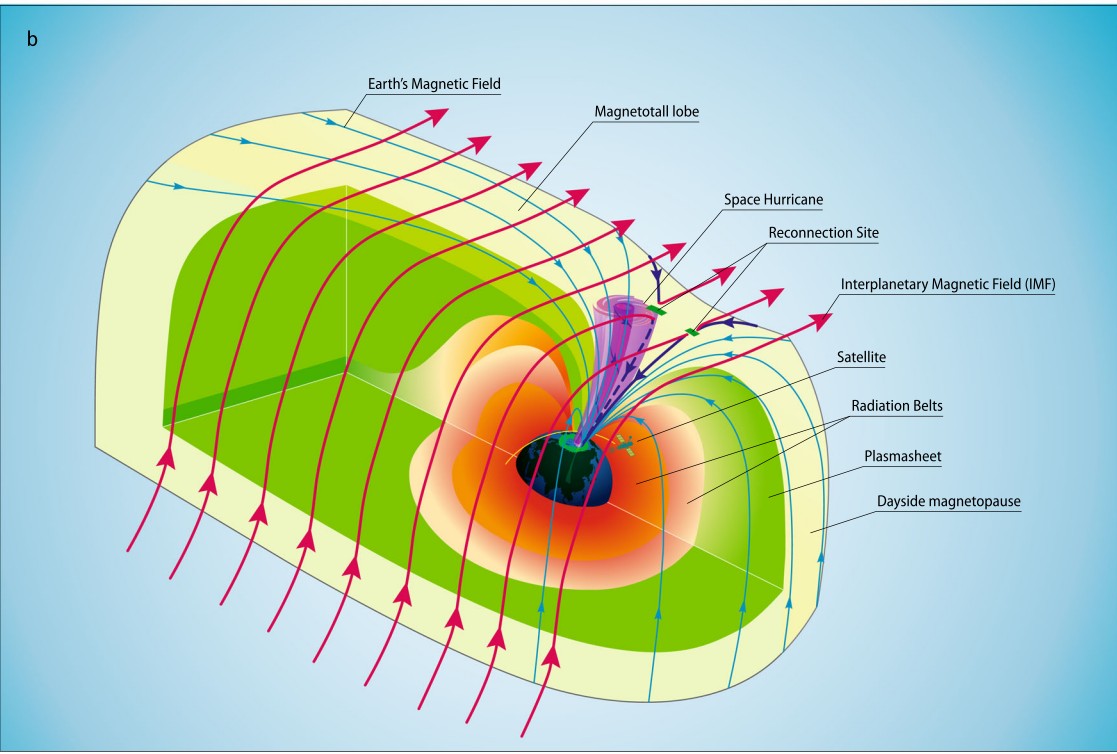

**Fig. 5 Schematic of the space hurricane and its formation mechanism during an extremely quiet geomagnetic condition with northward IMF and a dominant *By* component. a** Schematic of a space hurricane in the northern polar ionosphere. The magenta cyclone-shape auroral spot with brown thick arrows of circular ionospheric flows represents the space hurricane with a light green background showing the downward FACs. Convection streamlines are in blue with green thick crossed bars that shows the projected magnetic reconnection sites at the dayside magnetopause around equatorward and tailward (lobe) boundary of the cusp[29,30]. The vertical dark blue lines represent the Earth's magnetic field lines with electron precipitations and FACs. The sun is on the top representing the polar ionosphere is under sunlight conditions during the interval of interest. **b** Schematic of the 3-D magnetosphere when a space hurricane happened. Different color shadings represent different regions of the magnetosphere. The shaded magenta funnel shows the space hurricane in the magnetosphere. Red, black and blue curves with arrows are the interplanetary magnetic field lines, Earth's magnetic field lines, and newly reconnected Earth's magnetic field lines. The green thick bars represent the reconnection sites. The yellow curve with a satellite icon shows the satellite orbit. In this case, magnetopause reconnection can take place at the dayside magnetopause around equatorward and tailward (lobe) boundary of the cusp[29,30]. Due to a steady high-latitude lobe reconnection, a funnel (space hurricane) formed just poleward of the cusp region (**b**), and a large ionospheric convection lobe-cell with strong circular horizontal plasma flow inside the normal afternoon convection cell (**a**).

A Cartesian coordinate system has been used in the model with the Earth's center at the origin with X-axis pointing towards the Sun, Y-axis towards the dawn-dusk direction, and Z-axis towards the north. The size of the numerical box extends from 30 $R_E$ to –100 $R_E$ along the Sun-Earth line and from −50 $R_E$ to 50 $R_E$ in Y and Z directions, with $320 \times 320 \times 320$ grid points and a minimum grid spacing of 0.15 $R_E$. In order to avoid the complexities associated with the plasmasphere and large MHD characteristic velocity from the strong magnetic field, an inner boundary of radius 3 $R_E$ is set for the magnetosphere[24]. For allowing an electrostatic coupling process introduced between the ionosphere and the magnetospheric inner boundary, the model imbeds an electrostatic ionosphere shell with height-integrated conductance. An approximately dipole field has been used as the Earth's magnetic field with a dipole moment of $8.06 \times 10^{22}$ A/m in magnitude. For the current event, the model is run to solve the whole system by using the measured interplanetary conditions as inputs.

## Data availability

The THEMIS B solar wind and IMF data are available on http://themis.ssl.berkeley.edu/data/themis/thb/l2/esa/ and http://themis.ssl.berkeley.edu/data/themis/thb/l2/fgm/, respectively. The SYM-H and AE indices data is available on http://wdc.kugi.kyoto-u.ac.jp/dstdir/. The DMSP SSUSI and particle data is available on https://ssusi.jhuapl.edu/gal_AUR, http://sd-www.jhuapl.edu/Aurora/ and ftp://ftp.ngdc.noaa.gov/STP/satellite_data/, respectively. The AMPERE field-aligned current is available on http://ampere.jhuapl.edu/rBrowse/index.html. The 3D PPMLR-MHD simulation data is available on https://doi.org/10.5281/zenodo.4395721 with a separate DOI of https://doi.org/10.5281/zenodo.4395721.

## Code availability

The computer code of PPMLR-MHD model for simulating the formation of space hurricane is a large simulation program system, which need to be run on a supercomputer and will be available upon request to the contributed author (Chi Wang, cw@spaceweather.ac.cn).

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

## Acknowledgements

The work was supported by the National Natural Science Foundation of China (Grants 41874170, 41574138, 41604139, and 41831072), the Chinese Meridian Project, the Specialized Research Fund for State Key Laboratories of China and National Key Laboratory of Electromagnetic Environment, Research Councils United Kingdom Science and Technology Facilities Council Consolidated Grant ST/M000885/1, and the Research Council of Norway Grants 223252 and 230996. The computations were performed by Numerical Forecast Modeling Research & Development and Virtual Reality

System of State Key Laboratory of Space Weather. We acknowledge NASA Contract NAS5-02099 and V. Angelopoulos for the use of data from the THEMIS Mission, the World Data Center for Geomagnetism, Kyoto for making available the SYM-H and AE indices data, and the JHU/APL and NOAA for making available the DMSP data. The authors also thank the International Space Science Institute (ISSI/ISSI-BJ) for supporting workshops of our international team entitled Multi-Scale Magnetosphere-Ionosphere-Thermosphere Interaction.

## Author contributions

Q.H.Z. conceived the idea, collected data and wrote the manuscript. Y.L.Z. contributed to conceive the idea and is responsible for verification of the DMSP SSUSI data. C.W. and B. B.T. ran the simulation model and were involved in the scientific discussion. Y.Z.M. and X.Y.W. processed the DMSP plasma data. K.O., L.R.L., M.L., H.G.Y., J.I.M., Z.Y.X., Y.F. N. and L.D.X. participated in the scientific discussions and paper revision. All authors discussed the results and commented on the manuscript.

## Competing interests

The authors declare no competing interests.
