## [Peer Review File · Nature Communications]

Reviewers' comments:

Reviewer #1 (Remarks to the Author):

This paper presents observations of a polar auroral structure associated with field-aligned currents. The feature resembles a swirling meteorological hurricane with spiral arms and a stagnation point at its center, which the authors use as motivation to call the structure a 'space hurricane.' Observations of precipitating electrons and a secondary convection cell inferred from the velocity profile suggest the space hurricane is the result of high-latitude lobe reconnection. The authors also conducted global magnetosphere simulations and compared the calculated field-aligned currents to the observations. Finally, the authors provide a schematic explaining the formation of the space hurricane. While the observations are interesting and the results and discussion are detailed, there is some logic that is difficult to follow, particularly in relation to the different direction of rotation of the hurricane with respect to the ionospheric flow, and also to understand the geometry of the draping that drives the circulation of flux in the field-aligned current funnel (please see my detailed suggestions below). I recommend the paper be returned to the authors and revised as follows.

Line 26: Change "However, disturbance, resembling hurricanes, have" to "However, disturbances resembling hurricanes have"

Line 63: Change "nearly half time" to "nearly half of the time"

Line 94: Change "strong and dominated northward IMF" to "strongly northward dominated IMF"

Line 100: "the auroral oval (between 70 and 85 MLAT) was very weak except for auroral arcs in the dusk sector"

I find this to be slightly misleading. While there are almost no emissions coming from the dawn sector, I would not say it is fair to call the auroral oval weak when there are strong emissions, which you mention as the dusk arcs, coming from almost half of the portion of the oval that is visible in Figure 1 and Movie 1. Rather I would suggest something similar to "the auroral oval (between 70 and 85 MLAT) was generally quiet in the dawn sector while strong arcs persisted in the dusk sector."

Line 112: "the horizontal ionospheric plasma drift"

Please specify more clearly what direction is meant by 'horizontal.'

Line 116: “these flow shears give a clockwise circulation of ionospheric flow, which is opposite to the rotation trend of the auroral spot, indicating they are generated by dynamical processes”

This is the puzzling aspect of these observations and is also related to my comment about Line 155 below. Although you provide an explanation for this later, here is a wild thought: the hurricane is in fact rotating in the same direction as the ionospheric flow, but we are seeing the same effect as the rims of cars that appear to be spinning backwards on television. I have never used the SSUSI instrument myself but I found on this website

https://ssusi.jhuapl.edu/instrument_system_description that the time between successive images is typically 15-20 minutes, which is way longer than the timescale on which auroras can move and change. It seems possible that the hurricane could be spinning in the same direction as the ionospheric flows but with a rotation period slightly less than the time between SSSI observations. Would this explanation still be consistent with the rest of your observations?

Line 124: “These observations indicate that the space hurricane contains accelerated electron precipitation from the high-latitude lobe region of the magnetosphere”

While I understand the 'inverted-V' structures indicate 'accelerated electron precipitation' it is not clear to me why the source is necessarily the 'high-latitude lobe region of the magnetosphere.' Because of the lack of conjugate auroral signature I think this is a likely scenario but not a definite source. I would suggest adding “likely sourced” so it reads “These observations indicate that the space hurricane contains accelerated electron precipitation likely sourced from the high-latitude lobe region of the magnetosphere”

Line 138: “In the center of Figure 3a, a strong upward FAC funnel appears to nearly link to the inner edge of the high-latitude magnetopause FAC belt from the polar ionosphere (Figure 3b) surrounding and close with a strong downward FAC band at its morning side.”

While I do understand the content of this it took me many many times of reading it to do so. I suggest rewording into 2 sentences as such “In the center of Figure 3a, a strong upward FAC funnel appears to nearly link to the inner edge of the high-latitude magnetopause FAC belt from the polar ionosphere. Figure 3b shows the upward FAC closing through a strong downward FAC band on the dawn side.”

Line 140: “The funnel of FAC appears as a spot with several 'arms' and a trend of clockwise rotation (Movie S3)”

As I watch the movie I see that the vector field shows a trend of clockwise rotation throughout the whole magnetosphere. Are you suggesting the rotation of the arms is just being advected with this large scale flow pattern or is there an additional source of vorticity?

Line 140: "The funnel of FAC appears as a spot with several 'arms' and a trend of clockwise rotation (Movie S3), confirming the auroral observations and suggesting it is generated by high-latitude lobe reconnections."

How can I conclude from "The funnel of FAC appears as a spot with several 'arms' and a trend of clockwise rotation" that "it is generated by high-latitude lobe reconnection," these two don't seem to immediately follow.

Line 143: Change "(probably through the Knight's current-voltage process)" to "(probably through the Knight current-voltage process)"

Line 146: Change "These consistencies confirm" to "These consistencies provide strong evidence"

Line 155: "The newly reconnected open field lines are draped by the solar wind to move sunward and then tailward from the morning side to the afternoon side in the high-latitude lobe region."

It is not clear to me how the geometry of the draping leads to sunward movement. I was hoping Figure 4b would help understand this draping but it does not seem to indicate any sunward movement of newly reconnected field lines. Would the authors please provide some schematic description or a reference to understand this motion?

Line 165: Change "form a cyclone-shaped funnel due to the pressure gradient (see Figure 4b)" to "form a cyclone-shaped funnel (see Figure 4b) due to the pressure gradient"

Line 167: "This funnel becomes the most efficient channel to transfer the solar wind energy and mass" I think this is the most important statement of the manuscript and should really be the focus of the study. It would be very interesting if you could quantify the mass and/or energy transport. Can you provide quantitative comparisons with some other processes also occurring during quiet conditions, like, for instance, diffusive transport (Johnson and Cheng, 1997)? As far as I know the amount of mass transported by high-latitude lobe reconnection has only been quantified in case studies, so the present study would be an interesting case with complementary auroral observations.

Line 328 Supplementary Information: Change “The PPMLR-MHD model is on the basis of an extension” to “The PPMLR-MHD model is built on the basis of an extension”

Movies 1 and 2: I very much like Movies 1 and 2 provided for panels a and b of Figure 2, however it would help the reader to compare the content of the different animations if both were contained in the same movie, similarly to the way panels a and b are arranged in Figure 2.

Comment on Movies: Please provide animations in a format that can be paused. Without this it is almost impossible to identify some of the features in the movies, especially at the present frame rate.

Figure 4: Why are there 2 reconnection sites?

Line 300 Figure 4 caption: “Convection streamlines are in blue with green crossed bars”

What is meant by 'crossed bars?'

Brandon Burkholder

Reviewer #2 (Remarks to the Author):

Review of the manuscript entitled "A Space Hurricane over the Earth's Polar Ionosphere" by Zhang et al., NCOMMS-20-05941-T

The manuscript describes a phenomenon of intense auroral precipitation in the center of horizontal plasma flow. The authors draw connections to the appearance of hurricanes in the low atmosphere and call their event a “space hurricane”. I cannot find enough new information in this manuscript compared to prior publications about similar events and therefore I cannot recommend publication in Nature Communications. Detailed comments will be given below. Also, it will be very easy for the

authors to guess who the reviewer might have been and therefore I will sign the review with my name.

The most important criticism on this manuscript is that it does not give any quantitative estimate of the total mass and energy transport in their event. The manuscript does not compare the event to an average substorm or a geomagnetic storm. Such a comparison is required to justify the naming of a “space hurricane”. See lines 71-74.

Many of the phenomenon details have already been described earlier. The authors cite Frey, H. U. et al., Properties of localized, high latitude, dayside aurora, *J. Geophys. Res.* 108(A4), 8008 (2003) which I will call Frey-2003 but they miss two other publications Frey et al., Seasonal dependence of localized, High Latitude Dayside Aurora (HiLDA), *J. Geophys. Res.*, 109, A04303, 10.1029/2003JA010293, 2004 and Frey, H.U., Localized aurora beyond the auroral oval, *Rev. Geophys.*, 45, RG1003, 10.1029/2005RG000174, 2007, which I will call Frey-2004 and Frey-2007, respectively.

Line 55: The authors refer to earlier observations with electrons of 1 keV, much lower than their observation (Line 121), but actually Figures 8 and 11 of Frey-2003 show FAST measurements of up to 10 keV over the HiLDA spots (see also top right column of page 5 of Frey-2003).

Line 95: The appearance of the spot after the drop in solar wind density with dominating positive IMF B_y is described in Frey-2003.

Lines 96, 124: It has been speculated that the appearance of such a spot in the southern hemisphere required a negative IMF B_y (Frey-2007) and this has actually recently be confirmed by Carter et al., *JGR*, 2018. Frey-2004 also describes this as a summer phenomenon and therefore there cannot be a simultaneous spot in the southern winter hemisphere.

Line 99: The disappearance of the spot with a southward turn of the IMF has been described in Frey-2003.

Line 106: The colocation of the upward FAC and spot to a downward FAC into the cusp is described in Frey-2003.

Line 116: The clockwise convection cell with the spot is described in Frey-2003. Model calculations are also presented in Figure-6 of Frey-2004.

Line 117: What do the authors mean with “generated by dynamical processes”. That statement needs more explanation.

Line 123: The absence of ion precipitation is described in Frey-2003 and Frey-2007.

Lines 126-127: This statement requires references.

Lines 140-142: The authors claim that the spot is generated by high-latitude lobe reconnection. That is in contradiction to the mapping in Figure-12 of Frey-2003 which shows the connection between high-latitude magnetopause reconnection and the downward current into the cusp, while the upward current of the spot maps far down into the tail.

Here I also want to refer to lines 38-40 in the abstract that states “This study shows rapid transferring of solar wind energy and mass into the Earth’s magnetosphere...”. Which evidence do the authors have for the rapid and direct transfer of mass from the solar wind into their “hurricane”? Frey-2003 and Frey-2007 make the connection between the spots and regions far down in the magnetotail which would rather point to a delayed transfer of energy from the magnetosphere and not from the solar wind.

Lines 142-145: This electron acceleration is described in Frey-2004.

Lines 164-177: This process is shown schematically in Figure-39 of Kuijpers et al., Electric current circuits in astrophysics, Space Sci. Rev., 188, 3-57, 10.1007/s11214-014-0041-y, 2015. In astrophysics it is called “current-starvation”.

Lines 184-187: These statements are given without supporting references.

Figure 3: Wouldn’t it make sense to show the field aligned current plot in the same orientation as the others, with the sun to the right?

Figure 4b: I am totally surprised by the statement in the figure caption “In this case, magnetopause reconnection can take place at both the dayside low- and high-latitude (lobe) magnetopause.” How is that possible during northward IMF? What evidence do you have for that statement? This certainly needs more explanation.

This review was generated by Harald Frey, University of California, Berkeley.

Reviewer #3 (Remarks to the Author):

The paper presents an auroral spot observed by the DMSP satellite in the center of the polar cap that is accompanied by fast plasma flow around the aurora observed also by DMSP. Using the AMPERE satellites, the authors showed evidence for the upward field-aligned currents (FACs) above the auroral spot and downward FACs close to the aurora on the dayside, which intensified the plasma flow in the region bounded by the pair of FACs. Furthermore, the plasma flow is fast (2 km/s) for the quiet solar and geomagnetic conditions, which indicates energy input to the magnetosphere and ionosphere. The authors carried out global MHD simulations to confirm the fast plasma flow and FACs and concluded that the “Hurricane” like plasma vortex continued to exist in the center of the polar cap.

The data and simulations are solid, but I would like to raise several questions/comments. My major concern is whether or not the auroral spot is surrounded by fast flow in most of local times, but I did not find evidence for the fast plasma flow in the dusk-midnight sector in the data and simulation. The data and simulation show that the fast plasma flow is along the conventional round cell, in the center of which the auroral spot was observed.

[Comments]

The AMPERE satellites observed the upward FAC in the center of the polar cap and downward FAC equatorward around noon (Figure 2b). The fast plasma flow observed by the DMSP satellite in the afternoon sector (1540MLT) (Figure 2c) is consistent with the observed FACs. Faster plasma flow would be observed around noon, if several satellites were orbiting over the polar cap, judging from the distribution of the downward FAC and its counterpart, upward FAC. From the satellite observations, the authors inferred that there must be hurricane-like flow vortex in the center of the polar cap. To achieve the circular motion of plasma, there should be fast flow on the opposite local time sectors, particularly in the dusk-midnight sector. However, no evidence for fast flow is given in these local times. The MHD simulations (Figure 3b) show that the convection pattern is typical for IMF $B_z > 0$ and $B_y > 0$ conditions, composed of the round cell on the duskside and the crescent cell on the dawnside. The FACs observed by AMPERE are consistent with the simulation, if the crescent cell is turned a bit toward the noon meridian. As the authors mention in lines 106-107, the upward FAC is closed by downward cusp FAC, which leads to that high speed flow should be limited in the dayside region bounded by the upward and downward FACs. Thus, the observed fast flow may be in the region bounded by the crescent cell and round cell. On the other hand, the AMPERE observations and the simulation indicate that the convection flow lines expand equatorward in the dusk-midnight sector, rather than being limited in a circular region. Thus, the plasma flow may not resemble the vortex as schematically shown in Figure 4a. The equatorward expansion of the clockwise vortex is not consistent with the schematic diagram in Figure 4a, where the clockwise "Hurricane" intrudes into the clockwise convection vortex. As a result, the clockwise motion is separated into two clockwise vortices: One is "Hurricane" and the other is longitudinally elongated convection in the evening sector. To achieve this pattern, there should be a sheet of downward FAC at the boundary between the two clockwise vortices. Otherwise, the flow vectors with opposite direction cancel each other at the front of the intrusion, which results in large scale convection in the dusk-midnight sector, similar to the round cell typical for the positive IMF B_y and positive IMF B_z . I would like to note that the equi-potential lines deduced from the AMPERE observations (Figure 2b) and simulations (Figure 3b) expand equatorward in the dusk-midnight sector, which does not meet the separated vortices depicted in Figure 4a.

In summary, the DMSP satellite detected clockwise flow in the afternoon and dawn sectors, which is consistent with the AMPERE observation of the FACs. The auroral spot detected by DMSP is also in favor of the upward FAC in the center of the polar cap. However, there is no convincing data in the dusk-midnight sector that supports the circular plasma convection looking like "Hurricane". The MHD simulation well reproduced the upward FAC in the center of the polar cap and downward FACs in the dawn sector. But, the reproduced circulation of ionospheric plasma extends equatorward in the dusk-midnight sector, so that the pattern looks like the conventional round cell. I would like to conclude that the data is not enough to verify the "Hurricane".

[Minor comments]

Figure 4a.

The plasma convection is associated with the radial electric field converging to the upward FAC, which does not allow the radial flow of plasma.

Figure 4b

Please make clear the relationship between the reconnection site and the generator that supplies the field-aligned currents. The reconnection may have created magnetic stresses on one side of the funnel, while there should be magnetic stresses on the other side of the funnel to complete the FACs.

Line 90.

The “anti-clockwise” is not consistent with the clockwise flow in Figure 4a.

Lines 175-176

The field-aligned electric field is needed to accelerate electrons, but the FAC is generated by a dynamo or generator in the magnetosphere, which may be activated by the lobe reconnection. Since the FACs are well reproduced by the simulation code employed in the present paper, we could have an idea about the dynamo.

Takashi Kikuchi

Response to Reviewer #1

We thank Dr. Brandon Burkholder for many positive and constructive comments. We have carefully revised our manuscript based on your comments and suggestions. Please find below our replies to each of the comments.

This paper presents observations of a polar auroral structure associated with field-aligned currents. The feature resembles a swirling meteorological hurricane with spiral arms and a stagnation point at its center, which the authors use as motivation to call the structure a 'space hurricane.' Observations of precipitating electrons and a secondary convection cell inferred from the velocity profile suggest the space hurricane is the result of high-latitude lobe reconnection. The authors also conducted global magnetosphere simulations and compared the calculated field-aligned currents to the observations. Finally, the authors provide a schematic explaining the formation of the space hurricane. While the observations are interesting and the results and discussion are detailed, there is some logic that is difficult to follow, particularly in relation to the different direction of rotation of the hurricane with respect to the ionospheric flow, and also to understand the geometry of the draping that drives the circulation of flux in the field-aligned current funnel (please see my detailed suggestions below). I recommend the paper be returned to the authors and revised as follows.

We thank the reviewer for his/her appreciation on the potential contribution of our manuscript.

Line 26: Change “However, disturbance, resembling hurricanes, have” to “However, disturbances resembling hurricanes have”

Reply: Yes, we have improved this by following your suggestion.

Line 63: Change “nearly half time” to “nearly half of the time”

Reply: Yes, we have improved this by following your suggestion.

Line 94: Change “strong and dominated northward IMF” to “strongly northward dominated IMF”

Reply: Yes, we have improved this by following your suggestion.

Line 100: “the auroral oval (between 70 and 85 MLAT) was very weak except for auroral arcs in the dusk sector”

I find this to be slightly misleading. While there are almost no emissions coming from the dawn sector, I would not say it is fair to call the auroral oval weak when there are

strong emissions, which you mention as the dusk arcs, coming from almost half of the portion of the oval that is visible in Figure 1 and Movie 1. Rather I would suggest something similar to “the auroral auroral (between 70 and 85 MLAT) was generally quiet in the dawn sector while strong arcs persisted in the dusk sector.”

Reply: Yes, we have improved the sentence by following your suggestion.

Line 112: “the horizontal ionospheric plasma drift”

Please specify more clearly what direction is meant by 'horizontal.'

Reply: Yes, we have improved this by following your suggestion.

Line 116: “these flow shears give a clockwise circulation of ionospheric flow, which is opposite to the rotation trend of the auroral spot, indicating they are generated by dynamical processes”

This is the puzzling aspect of these observations and is also related to my comment about Line 155 below. Although you provide an explanation for this later, here is a wild thought: the hurricane is in fact rotating in the same direction as the ionospheric flow, but we are seeing the same effect as the rims of cars that appear to be spinning backwards on television. I have never used the SSUSI instrument myself but I found on this website https://ssusi.jhuapl.edu/instrument_system_description that the time between successive images is typically 15-20 minutes, which is way longer than the timescale on which auroras can move and change. It seems possible that the hurricane could be spinning in the same direction as the ionospheric flows but with a rotation period slightly less than the time between SUSSI observations. Would this explanation still be consistent with the rest of your observations?

Reply: We thank you for your suggestions, but we think it may be hard to explain the opposite trend between the cyclone-shape aurora and circular ionospheric flow by using your wild thought, because the aurora arms are formed by particle precipitation and represents the trace of reconnected magnetic field lines. This is confirmed by the clockwise motions of the reconnected magnetic field lines by a roughly steady lobe magnetic reconnections from our simulations. These motions result in a funnel formed for large and rapid deposition of solar wind/magnetosphere energy and momentum into the ionosphere around the magnetic pole, contrary to the established expectation during extremely quiet geomagnetic conditions. Thus, it may be different from your “wild thought: the rims of cars that appear to be spinning backwards on television”. We added the following sentences:

“...Inside the funnel, a corkscrew magnetic field forms with circular flow and upward FACs, which accelerate electrons that precipitate into the ionosphere^{15,23,30} and create the auroral spot with multiple “arms” as observed by DMSP F16 in Figure 2a. In other words, the auroral “arms” represent the trace of the footprints of the reconnected

magnetic field lines, and shows an illusionary trend of anti-clockwise rotation, which is opposite to the flow circulation and different from tropospheric hurricanes.

Line 124: “These observations indicate that the space hurricane contains accelerated electron precipitation from the high-latitude lobe region of the magnetosphere”

While I understand the 'inverted-V' structures indicate 'accelerated electron precipitation' it is not clear to me why the source is necessarily the 'high-latitude lobe region of the magnetosphere.' Because of the lack of conjugate auroral signature I think this is a likely scenario but not a definite source. I would suggest adding “likely sourced” so it reads “These observations indicate that the space hurricane contains accelerated electron precipitation likely sourced from the high-latitude lobe region of the magnetosphere”

Reply: Yes, we have improved this by following your suggestion.

Line 138: “In the center of Figure 3a, a strong upward FAC funnel appears to nearly link to the inner edge of the high-latitude magnetopause FAC belt from the polar ionosphere (Figure 3b) surrounding and close with a strong downward FAC band at its morning side.”

While I do understand the content of this it took me many many times of reading it to do so. I suggest rewording into 2 sentences as such “In the center of Figure 3a, a strong upward FAC funnel appears to nearly link to the inner edge of the high-latitude magnetopause FAC belt from the polar ionosphere. Figure 3b shows the upward FAC closing through a strong downward FAC band on the dawn side.”

Reply: Yes, we have reworded this sentence into two sentences by following your suggestion.

Line 140: “The funnel of FAC appears as a spot with several 'arms' and a trend of clockwise rotation (Movie S3)”

As I watch the movie I see that the vector field shows a trend of clockwise rotation throughout the whole magnetosphere. Are you suggesting the rotation of the arms is just being advected with this large scale flow pattern or is there an additional source of vorticity?

Reply: Yes, we suggest the rotation of the arms is just being advected with the large-scale flow pattern.

Line 140: “The funnel of FAC appears as a spot with several 'arms' and a trend of clockwise rotation (Movie S3), confirming the auroral observations and suggesting it is generated by high-latitude lobe reconnections.”

How can I conclude from “The funnel of FAC appears as a spot with several 'arms'”

and a trend of clockwise rotation” that “it is generated by high-latitude lobe reconnection,” these two don't seem to immediately follow.

Reply: We find the direct evidence of steady high latitude lobe magnetic reconnection from 3-D distribution of the simulated magnetic field lines, see in Figure 3b, Figure S2 and Movie S4.

Figure S2. Two different views of the 3-D selected magnetic field lines simulated by the PPMLR-MHD code for the snapshot shown in Figure 3. The format is the same as Figure 3b.

Line 143: Change “(probably through the Knight's current-voltage process)” to “(probably through the Knight current-voltage process)”

Reply: Yes, we have improved this by following your suggestion.

Line 146: Change “These consistencies confirm” to “These consistencies provide strong evidence”

Reply: Yes, we have improved this by following your suggestion.

Line 155: “The newly reconnected open field lines are draped by the solar wind to move sunward and then tailward from the morning side to the afternoon side in the high-latitude lobe region.”

It is not clear to me how the geometry of the draping leads to sunward movement. I was hoping Figure 4b would help understand this draping but it does not seem to indicate any sunward movement of newly reconnected field lines. Would the authors please provide some schematic description or a reference to understand this motion?

Reply: We thank you for pointing out this for us. Following your suggestions, we have added a reference which discusses the evolution of magnetic field lines after

high-latitude lobe reconnection during northward IMF (Lockwood and Moen, 1999). We have revised the text as follows:

“The newly reconnected open field lines are draped by the solar wind to move dawnward and then tailward from the morning side to the afternoon side in the high-latitude lobe region²¹.”

Line 165: Change “form a cyclone-shaped funnel due to the pressure gradient (see Figure 4b)” to “form a cyclone-shaped funnel (see Figure 4b) due to the pressure gradient”

Reply: Yes, we have improved this by following your suggestion.

Line 167: “This funnel becomes the most efficient channel to transfer the solar wind energy and mass” I think this is the most important statement of the manuscript and should really be the focus of the study. It would be very interesting if you could quantify the mass and/or energy transport. Can you provide quantitative comparisons with some other processes also occurring during quiet conditions, like, for instance, diffusive transport (Johnson and Cheng, 1997)? As far as I know the amount of mass transported by high-latitude lobe reconnection has only be quantified in case studies, so the present study would be an interesting case with complementary auroral observations.

Reply: Yes, we thank you for your nice suggestion. We have now made the detailed calculation of the energy and mass transfer and found the total precipitating electron energy flux (ΣJE) over the space hurricane is ~90% of the ΣJE over the entire polar pass of DMSP F16, which is about 10 times higher than that during a polar pass of a general quiet condition, and is only about 4.6 times less than that over a polar pass in the main phase of the first super geomagnetic storm of solar cycle 24 (but the duration of the entire polar pass with the space hurricane is only one third of that for the super storm case, so that the pass with the space hurricane has 3.2 times higher average energy flux than that for the super storm case). We added a table and a paragraph to show the details:

“...Within the space hurricane, the total energy flux (JE) and the average energy of the precipitating electrons were significantly increased (Figure 2g and 2h), resulting in a time integrated JE (ΣJE) up to 2.48×10^{14} eV/(cm²·sr) from 16:16:58 to 16:18:51 UT, which is about 91.49% of the ΣJE (2.71×10^{14} eV/(cm²·sr)) for the whole polar pass (see Table 1). The ΣJE (2.71×10^{14} eV/(cm²·sr)) is about 10 times higher than that for a polar pass without a space hurricane also under a geomagnetic quiet condition (see Table 1 for the DMSP pass from 08:54:22 to 09:11:24 UT on 21 June 2010). Furthermore, it is only about 4.6 times smaller than the ΣJE of a pass during the main phase of the first super geomagnetic storm of solar cycle 24 which

had intense solar wind driving and strong southward IMF (see Table 1 for the pass from 23:14:00 to 23:44:00 UT on 17 March 2015). The space hurricane has an average energy flux about 5.5 times higher than its whole polar pass shown in Figure 2, about 15.1 times higher than the typical quiet case, and even 3.2 times higher than the super storm case (see Table 1). Table 1 also shows that the average energy flux in the space hurricane (2.2×10^{12} eV/(cm²·s·sr)) is about 28.8 times higher than that in the auroral oval (7.8×10^{10} eV/(cm²·s·sr)) and 71.7 times higher than that in the diffuse aurora region (3.1×10^{10} eV/(cm²·s·sr)) during the same pass (see Figure 2h). The space hurricane also has the highest maximum and average energy in the magnetic pole region compared to the values during typical quiet and super storm times in the same region (see Table 1). These indicate that the space hurricane leads to large and rapid deposition of energy and flux into the polar ionosphere during an otherwise extremely quiet geomagnetic condition, suggesting that current geomagnetic activity indicators do not properly represent the dramatic activity within space hurricanes, which are located further poleward than geomagnetic index observatories....”

Line 328 Supplementary Information: Change “The PPMLR-MHD model is on the basis of an extension” to “The PPMLR-MHD model is built on the basis of an extension”

Reply: Yes, we have improved this by following your suggestion.

Movies 1 and 2: I very much like Movies 1 and 2 provided for panels a and b of Figure 2, however it would help the reader to compare the content of the different animations if both were contained in the same movie, similarly to the way panels a and b are arranged in Figure 2.

Reply: Yes, we thank you for your nice suggestion. We have kept Movie S1 for showing the conjugate aurora in the northern and southern hemispheres, and we have overlapped the DMSP orbit with cross-track horizontal plasma flows in each auroral image of the top panel of Movie S1, and plotted it in the new Movie S2 together with the AMPERE FAC maps in old Movie S2 as the same format as the panels a and b of new Figure 2 by following your suggestions.

Comment on Movies: Please provide animations in a format that can be paused. Without this it is almost impossible to identify some of the features in the movies, especially at the present frame rate.

Reply: Yes, we thank you for your nice suggestion. We plotted all the movies in .avi files, which can be easily paused as you suggested.

Figure 4: Why are there 2 reconnection sites?

Reply: We thank you for pointing out this for us. Firstly, the idea of the “magnetopause reconnection can take place at both the dayside low- and high-latitude (lobe) magnetopause” is mainly based on the reconnection scenario proposed by Lockwood and Moen (1999), which suggested the magnetopause reconnection can take place at both the dayside low- (at the equatorward edge of the cusp) and high-latitude (lobe, at the poleward edge of the cusp) magnetopause. Secondly, we find the direct evidence of the reconnected magnetic field lines both at the equatorward and poleward (lobe) edges of the cusp from the simulations. Please see in Figure 3b, Figure S2 and Movie S4.

Figure S2. Two different views of the 3-D selected magnetic field lines simulated by the PPMLR-MHD code for the snapshot shown in Figure 3. The format is same as Figure 3b.

However, we now only mentioned the high-latitude lobe reconnection in the text because the magnetopause reconnection at the equatorward edge of the cusp does not contribute to the formation of space hurricane. We modified the statements in the text and figure caption as well as in Figure 4.

Line 300 Figure 4 caption: “Convection streamlines are in blue with green crossed bars” . What is meant by 'crossed bars'?

Reply: The crossed bar means the projected reconnection site in the polar ionosphere. We have made it more visible in the revised figure.

We thank you again for many positive and constructive comments and suggestions.

Response to Reviewer #2

We thank you, Dr. Harald Frey, for your valuable comments and concerns. We like to take this opportunity to discuss and share some different views.

First, following your and Dr. Burkholder's suggestions on making the quantitative estimation of the total mass and energy transport in our event, we calculated the energy flux and average energy from DMSP F16 particle and SSUSI image data, and found that the total electron energy flux (ΣJ_E) over the space hurricane is ~90% of the ΣJ_E over the entire polar pass of DMSP F16, which is about 10 times higher than that during a typical polar pass under a geomagnetic quiet condition without the space hurricane, and is only about 4.6 times less than that over the polar pass of a super storm case. More details are included in our following replies. These indicate that the space hurricane did lead to large and rapid transfer of energy flux into the polar ionosphere during this extremely quiet geomagnetic condition.

Second, the novelty of our work is on advancing the physical understanding on formation and evolution of space hurricanes by coordinated observations (hot precipitating particles, cold ionospheric plasma, and aurora) as well as 3D MHD simulations. Your previous studies (e.g. Frey-2003, 2004, and 2007) provided some valuable information on what we are calling a space hurricane, and you named it as High Latitude Dayside Aurora (HiLDA). The HiLDA studies were based on auroral images with a low spatial resolution. Ionospheric plasma velocity measurements were not included in your studies. Therefore, other important features of the phenomenon, (a cyclone-shaped aurora, a strong circular horizontal plasma flow with shears and a nearly zero-flow center), were not included in your studies. These new features, which we report in our manuscript, significantly advance our understanding of the space hurricane. To our knowledge, these new features have never been reported in the scientific literature. Thus, the current paper shows for the first time all key characteristics of the space hurricane, which resembles a hurricane in the lower atmosphere, and contradicts with our current understanding on how the solar wind - magnetosphere interact during periods of extremely low solar activity. Our study, therefore, not only identifies this previously unknown phenomena, but also provides a framework of new insight into the formation and evolution of the space hurricane. Furthermore, it shows how the space hurricane represents a fundamental process of

unexpected rapid transfer of solar wind mass and energy into the magnetosphere and ionosphere during periods of extremely low solar activity.

The manuscript describes a phenomenon of intense auroral precipitation in the center of horizontal plasma flow. The authors draw connections to the appearance of hurricanes in the low atmosphere and call their event a “space hurricane”. I cannot find enough new information in this manuscript compared to prior publications about similar events and therefore I cannot recommend publication in Nature Communications. Detailed comments will be given below. Also, it will be very easy for the authors to guess who the reviewer might have been and therefore I will sign the review with my name.

The most important criticism on this manuscript is that it does not give any quantitative estimate of the total mass and energy transport in their event. The manuscript does not compare the event to an average substorm or a geomagnetic storm. Such a comparison is required to justify the naming of a “space hurricane”. See lines 71-74.

Reply: Yes, we thank you for your nice suggestion. Firstly, we would like to point out that the space hurricane occurred during a geomagnetic quiet time, which suggests that it may be better to compare it with other typical quiet time phenomena. Secondly, we now made the detail calculation of the energy and mass transfer and found the total precipitating electron energy flux (ΣJE) over the space hurricane is ~90% of the ΣJE over the entire polar pass of DMSP F16, which is about 10 times higher than that during a polar pass of a general quiet condition, and is only about 4.6 times less than that over a polar pass in the main phase of the first super geomagnetic storm of solar cycle 24 (but the duration of the entire polar pass with the space hurricane is only one third of that for the super storm case, so that the pass with the space hurricane has 3.2 times higher average energy flux than that for the super storm case). We added a table and a paragraph to show the details:

“...Within the space hurricane, the total energy flux (JE) and the average energy of the precipitating electrons were significantly increased (Figure 2g and 2h), resulting in a time integrated JE (ΣJE) up to 2.48×10^{14} eV/(cm²·sr) from 16:16:58 to 16:18:51 UT, which is about 91.49% of the ΣJE (2.71×10^{14} eV/(cm²·sr)) for the whole polar pass (see Table 1). The ΣJE (2.71×10^{14} eV/(cm²·sr)) is about 10 times higher than that for a polar pass without a space hurricane also under a geomagnetic quiet condition (see Table 1 for the DMSP pass from 08:54:22 to 09:11:24 UT on 21 June 2010). Furthermore, it is only about 4.6 times smaller than the ΣJE of a pass during the main phase of the first super geomagnetic storm of solar cycle 24 which had intense solar wind driving and strong southward IMF (see Table 1 for the pass from 23:14:00 to 23:44:00 UT on 17 March 2015). The space hurricane has an

average energy flux about 5.5 times higher than its whole polar pass shown in Figure 2, about 15.1 times higher than the typical quiet case, and even 3.2 times higher than the super storm case (see Table 1). Table 1 also shows that the average energy flux in the space hurricane (2.2×10^{12} eV/(cm²·s·sr)) is about 28.8 times higher than that in the auroral oval (7.8×10^{10} eV/(cm²·s·sr)) and 71.7 times higher than that in the diffuse aurora region (3.1×10^{10} eV/(cm²·s·sr)) during the same pass (see Figure 2h). The space hurricane also has the highest maximum and average energy in the magnetic pole region compared to the values during typical quiet and super storm times in the same region (see Table 1). These indicate that the space hurricane leads to large and rapid deposition of energy and flux into the polar ionosphere during an otherwise extremely quiet geomagnetic condition, suggesting that current geomagnetic activity indicators do not properly represent the dramatic activity within space hurricanes, which are located further poleward than geomagnetic index observatories....”

Many of the phenomenon details have already been described earlier. The authors cite Frey, H. U. et al., Properties of localized, high latitude, dayside aurora, *J. Geophys. Res.* 108(A4), 8008 (2003) which I will call Frey-2003 but they miss two other publications Frey et al., Seasonal dependence of localized, High Latitude Dayside Aurora (HiLDA), *J. Geophys. Res.*, 109, A04303, 10.1029/2003JA010293, 2004 and Frey, H.U., Localized aurora beyond the auroral oval, *Rev. Geophys.*, 45, RG1003, 10.1029/2005RG000174, 2007, which I will call Frey-2004 and Frey-2007, respectively.

Reply: We have included all of these references and made some comparisons with them in the main text and we agree many of the features have been described earlier in the references. However, we would like to mention that the most important characteristics of the space hurricane, a cyclone-shaped aurora, a strong circular horizontal plasma flow with shears and a nearly zero-flow center shown in this paper, were missed in the HiLDA spots, which is likely due to a relatively low spatial resolution in the auroral image data and lack of coincident plasma drift measurement. We add the following sentences:

“...The observation features and formation conditions of the space hurricane are almost the same as the High Latitude Dayside Aurora (HiLDA) spots which were coordinated observed by IMAGE and FAST satellites¹²⁻¹⁶, indicating that the space hurricane and HiLDA spot may be the same phenomenon in the polar cap region. However, the most important characteristics of the space hurricane, a cyclone-shaped aurora, a strong circular horizontal plasma flow with shears and a nearly zero-flow center shown in this paper, were missed in the HiLDA spots based on previous

observations¹²⁻¹⁶, which are likely due to a relatively low spatial resolution in the auroral image data and lack of coincident ionospheric plasma drift measurement...”

Line 55: The authors refer to earlier observations with electrons of 1 keV, much lower than their observation (Line 121), but actually Figures 8 and 11 of Frey-2003 show FAST measurements of up to 10 keV over the HiLDA spots (see also top right column of page 5 of Frey-2003).

Reply: We thank you for reminding us about this. We now removed the statement of “(around 1keV)” after “precipitation electrons” for the HiLDA spots.

Line 95: The appearance of the spot after the drop in solar wind density with dominating positive IMF By is described in Frey-2003.

Reply: We now added the following sentence and cited the reference of Frey-2003:

“...with comparable solar wind number density ($N_{sw} = \sim 4 \text{ cm}^{-3}$) to the conditions described above (see Movie S1 and Figure 1), similar with the conditions for the appearance of the HiLDA spots¹¹.”

Lines 96, 124: It has been speculated that the appearance of such a spot in the southern hemisphere required a negative IMF By (Frey-2007) and this has actually recently be confirmed by Carter et al., JGR, 2018. Frey-2004 also describes this as a summer phenomenon and therefore there cannot be a simultaneous spot in the southern winter hemisphere.

Reply: We now added the following sentence and cited the references of Frey-2007, Frey-2004, and Carter-2018:

“...There is no conjugate auroral spot in the Southern Hemisphere (Movie S1), as expected from the circulation of plasma within the polar cap ionosphere under strong IMF By conditions^{22, 4, 11, 14, 15}....

...Note that there is almost no ion precipitation in the space hurricane area (Figure 2j) and no conjugate auroral structure in the Southern Hemisphere, same as that in the HiLDA spots¹¹.”

Line 99: The disappearance of the spot with a southward turn of the IMF has been described in Frey-2003.

Reply: We now added the following sentence and cited the reference of Frey-2003:

“...The space hurricane lasted about 8 hours, gradually decayed and merged into the duskside auroral oval around 20:00 UT when the IMF turned southward (see Movie S1 and Figure 1), same as the HiLDA spots disappears¹¹.”

Line 106: The collocation of the upward FAC and spot to a downward FAC into the

cusps is described in Frey-2003.

Reply: We now added the following sentence and cited the reference of Frey-2003: “...also shows a spot-like strong upward FAC (red, reaching above $1.5 \mu\text{Am}^{-2}$) within a negative potential cell (contours in Figure 2b), which is co-located with the space hurricane and also similar with that in HiLDA spots¹¹.”

Line 116: The clockwise convection cell with the spot is described in Frey-2003. Model calculations are also presented in Figure-6 of Frey-2004.

Reply: Yes, we have read both of your papers. In that line, we just describe our observations first and cited your papers in the following discussion parts.

Line 117: What do the authors mean with “generated by dynamical processes”. That statement needs more explanation.

Reply: We thank both Dr. Frey and Dr. Burkholder for pointing out this. We now deleted the statement of “generated by dynamical processes”, and we have added the following sentences to provide more explanation.

“...Clear electron inverted-V acceleration appeared within the space hurricane with ~ 10 keV energy electron precipitation near the hurricane center and ~ 1 keV energy electron precipitation around the edge (Figure 2h and 2i), the amount of electron energization increasing with increasing upward FAC strength due to an increasing field-aligned potential drop. The stronger FACs occur near the hurricane center to maintain ionospheric current continuity in the presence of convergent ionospheric Pedersen currents caused by the curvature of the circular flow increasing towards the hurricane center, inferring that a FAC spot or funnel with circular fast flows appears in the electron source region....”

Line 123: The absence of ion precipitation is described in Frey-2003 and Frey-2007.

Reply: We added the following sentence and cited the references of Frey-2003 and Frey-2007:

“...Note that there is almost no ion precipitation in the space hurricane area (Figure 2j) and no conjugate auroral structure in the Southern Hemisphere, same as that in the HiLDA spots¹¹.”

Lines 126-127: This statement requires references.

Reply: We thank you for reminding us about this. We have found several space hurricane events, which show many similar features to our present case. We are working on them, but we do not yet have them ready for publication. Thus, we deleted this statement.

Lines 140-142: The authors claim that the spot is generated by high-latitude lobe reconnection. That is in contradiction to the mapping in Figure-12 of Frey-2003 which shows the connection between high-latitude magnetopause reconnection and the downward current into the cusp, while the upward current of the spot maps far down into the tail.

Reply: We have now modified our claims as the space hurricane is generated by a process of keeping current continuity which is triggered by a steady high-latitude lobe magnetic reconnection. From our simulations, we can find that a strong upward FAC funnel, associated with the space hurricane and the ampere FAC spot, appears to nearly link to the inner edge of the high-latitude magnetopause FAC belt from the polar ionosphere. Thus, we don't think the upward current of the space hurricane maps too far down into the tail.

Here I also want to refer to lines 38-40 in the abstract that states "This study shows rapid transferring of solar wind energy and mass into the Earth's magnetosphere...". Which evidence do the authors have for the rapid and direct transfer of mass from the solar wind into their "hurricane"? Frey-2003 and Frey-2007 make the connection between the spots and regions far down in the magnetotail which would rather point to a delayed transfer of energy from the magnetosphere and not from the solar wind.

Reply: We thank you for pointing out this for us. We now made the detail calculation of the energy and mass transfer and found the total precipitating electron energy flux (ΣJE) over the space hurricane is ~90% of the ΣJE over the entire polar pass of DMSP F16, which is about 10 times higher than that during a polar pass of a general quiet condition, and is only about 4.6 times less than that over a polar pass in the main phase of the first super geomagnetic storm of solar cycle 24 (but the duration of the entire polar pass with the space hurricane is only one third of that for the super storm case, so that the pass with the space hurricane has 3.2 times higher average energy flux than that for the super storm case). We added a table and a paragraph to show the details:

"...Within the space hurricane, the total energy flux (JE) and the average energy of the precipitating electrons were significantly increased (Figure 2g and 2h), resulting in a time integrated JE (ΣJE) up to 2.48×10^{14} eV/(cm²·sr) from 16:16:58 to 16:18:51 UT, which is about 91.49% of the ΣJE (2.71×10^{14} eV/(cm²·sr)) for the whole polar pass (see Table 1). The ΣJE (2.71×10^{14} eV/(cm²·sr)) is about 10 times higher than that for a polar pass without a space hurricane also under a geomagnetic quiet condition (see Table 1 for the DMSP pass from 08:54:22 to 09:11:24 UT on 21 June 2010). Furthermore, it is only about 4.6 times smaller than the ΣJE of a pass during the main phase of the first super geomagnetic storm of solar cycle 24 which had

intense solar wind driving and strong southward IMF (see Table 1 for the pass from 23:14:00 to 23:44:00 UT on 17 March 2015). The space hurricane has an average energy flux about 5.5 times higher than its whole polar pass shown in Figure 2, about 15.1 times higher than the typical quiet case, and even 3.2 times higher than the super storm case (see Table 1). Table 1 also shows that the average energy flux in the space hurricane (2.2×10^{12} eV/(cm²·s·sr)) is about 28.8 times higher than that in the auroral oval (7.8×10^{10} eV/(cm²·s·sr)) and 71.7 times higher than that in the diffuse aurora region (3.1×10^{10} eV/(cm²·s·sr)) during the same pass (see Figure 2h). The space hurricane also has the highest maximum and average energy in the magnetic pole region compared to the values during typical quiet and super storm times in the same region (see Table 1). These indicate that the space hurricane leads to large and rapid deposition of energy and flux into the polar ionosphere during an otherwise extremely quiet geomagnetic condition, suggesting that current geomagnetic activity indicators do not properly represent the dramatic activity within space hurricanes, which are located further poleward than geomagnetic index observatories....”

And we also modified our statement as follows:

“This study shows large and rapid deposition of solar wind/magnetosphere energy and mass into the ionosphere around the magnetic pole unlike expectations during extremely quiet geomagnetic conditions.”

Lines 142-145: This electron acceleration is described in Frey-2004.

Reply: Yes, we now cite the reference when we described the electron acceleration as follows:

“...These upward FACs (both from the simulation and observations) cause magnetic field-aligned acceleration of magnetospheric electrons (probably through the Knight current-voltage process to keep current continuity^{23,29,30,5,14}) that precipitate into the polar ionosphere and generate the hurricane structure in the aurora...”

Lines 164-177: This process is shown schematically in Figure-39 of Kuijpers et al., Electric current circuits in astrophysics, Space Sci. Rev., 188, 3-57, 10.1007/s11214-014-0041-y, 2015. In astrophysics it is called “current-starvation”.

Reply: We thank you for reminding us about this. We have cited the reference, but we would like to use the “current continuity” in our paper to make it easily understand for broad audiences and add the following sentences:

“...The observations and simulations reveal that the space hurricane is generated by a steady high-latitude lobe magnetic reconnection^{3, 4} and the current continuity⁵ during a several hour period of northward interplanetary magnetic field (IMF) and very low solar wind density and speed.....

...During their dawnward and tailward motion, an elongated FAC sheet forms due to the flow shear, and the magnetosheath ions precipitate into the cusp ionosphere along field lines to give the downward FACs (like traces of dropping sands from a moving hourglass). In order to maintain current continuity in the ionosphere, the system sets up an upward FAC with a parallel potential that accelerates the existing electrons into the ionosphere and creates an “arm” of the auroral spot^{5, 15} observed by DMSP F16 in Figure 2a and shown in Figure 4a...

These upward FACs (both from the simulation and observations) cause magnetic field-aligned acceleration of magnetospheric electrons (probably through the Knight current-voltage process to keep current continuity^{23,29,30,5,14}) that precipitate into the polar ionosphere and generate the hurricane structure in the aurora...”

Lines 184-187: These statements are given without supporting references.

Reply: We thank you for reminding us about this. We now added the references.

Figure 3: Wouldn't it make sense to show the field aligned current plot in the same orientation as the others, with the sun to the right?

Reply: Yes, we thank you for your nice suggestion. We have rotated Figure 3b to make it with the sun to the right same as the others.

Figure 4b: I am totally surprised by the statement in the figure caption “In this case, magnetopause reconnection can take place at both the dayside low- and high-latitude (lobe) magnetopause.” How is that possible during northward IMF? What evidence do you have for that statement? This certainly needs more explanation.

Reply: We thank you for pointing out this for us. Firstly, the idea of the “magnetopause reconnection can take place at both the dayside low- and high-latitude (lobe) magnetopause” is mainly based on the reconnection scenario proposed by Lockwood and Moen (1999), which suggested the magnetopause reconnection can take place at both the dayside low- (at the equatorward edge of the cusp) and high-latitude (lobe, at the poleward edge of the cusp) magnetopause. Secondly, we find the direct evidence of the reconnected magnetic field lines both at the equatorward and poleward (lobe) edges of the cusp from the simulations. Please see in Figure 3b, Figure S2 and Movie S4.

Figure S2. Two different views of the 3-D selected magnetic field lines simulated by the PPMLR-MHD code for the snapshot shown in Figure 3. The format is same as Figure 3b.

However, we now only mentioned the high-latitude lobe reconnection in the text because the magnetopause reconnection at the equatorward edge of the cusp does not contribute to the formation of space hurricane. We modified the statements in the text and figure caption as well as in Figure 4.

We thank you again for many valuable comments, suggestions and especially these studies on HiLDA.

Response to Reviewer #3

We thank Dr. Takashi Kikuchi for your positive and constructive comments on our paper. Your comments and suggestions have been fully taken into account during revision. Below we provide point-by-point responses.

The paper presents an auroral spot observed by the DMSP satellite in the center of the polar cap that is accompanied by fast plasma flow around the aurora observed also by DMSP. Using the AMPERE satellites, the authors showed evidence for the upward field-aligned currents (FACs) above the auroral spot and downward FACs close to the aurora on the dayside, which intensified the plasma flow in the region bounded by the

pair of FACs. Furthermore, the plasma flow is fast (2 km/s) for the quiet solar and geomagnetic conditions, which indicates energy input to the magnetosphere and ionosphere. The authors carried out global MHD simulations to confirm the fast plasma flow and FACs and concluded that the “Hurricane” like plasma vortex continued to exist in the center of the polar cap.

The data and simulations are solid, but I would like to raise several questions/comments. My major concern is whether or not the auroral spot is surrounded by fast flow in most of local times, but I did not find evidence for the fast plasma flow in the dusk-midnight sector in the data and simulation. The data and simulation show that the fast plasma flow is along the conventional round cell, in the center of which the auroral spot was observed.

Reply: We thank you for thinking our data and simulations are solid and pointing out that we need further evidence of the auroral spot surrounded by fast flow. We will respond to that in the comments below.

[Comments]

The AMPERE satellites observed the upward FAC in the center of the polar cap and downward FAC equatorward around noon (Figure 2b). The fast plasma flow observed by the DMSP satellite in the afternoon sector (1540MLT) (Figure 2c) is consistent with the observed FACs. Faster plasma flow would be observed around noon, if several satellites were orbiting over the polar cap, judging from the distribution of the downward FAC and its counterpart, upward FAC. From the satellite observations, the authors inferred that there must be hurricane-like flow vortex in the center of the polar cap. To achieve the circular motion of plasma, there should be fast flow on the opposite local time sectors, particularly in the dusk-midnight sector. However, no evidence for fast flow is given in these local times. The MHD simulations (Figure 3b) show that the convection pattern is typical for IMF $B_z > 0$ and $B_y > 0$ conditions, composed of the round cell on the duskside and the crescent cell on the dawnside. The FACs observed by AMPERE are consistent with the simulation, if the crescent cell is turned a bit toward the noon meridian. As the authors mention in lines 106-107, the upward FAC is closed by downward cusp FAC, which leads to that high speed flow should be limited in the dayside region bounded by the upward and downward FACs. Thus, the observed fast flow may be in the region bounded by the crescent cell and round cell. On the other hand, the AMPERE observations and the simulation indicate that the convection flow lines expand equatorward in the dusk-midnight sector, rather than being limited in a circular region. Thus, the plasma flow may not resemble the vortex as schematically shown in Figure 4a. The equatorward expansion of the clockwise vortex is not consistent with the schematic diagram in Figure 4a, where the clockwise “Hurricane” intrudes into the clockwise convection vortex. As a result, the

clockwise motion is separated into two clockwise vortices: One is “Hurricane” and the other is longitudinally elongated convection in the evening sector. To achieve this pattern, there should be a sheet of downward FAC at the boundary between the two clockwise vortices. Otherwise, the flow vectors with opposite direction cancel each other at the front of the intrusion, which results in large scale convection in the dusk-midnight sector, similar to the round cell typical for the positive IMF B_y and positive IMF B_z . I would like to note that the equi-potential lines deduced from the AMPERE observations (Figure 2b) and simulations (Figure 3b) expand equatorward in the dusk-midnight sector, which does not meet the separated vortices depicted in Figure 4a.

In summary, the DMSP satellite detected clockwise flow in the afternoon and dawn sectors, which is consistent with the AMPERE observation of the FACs. The auroral spot detected by DMSP is also in favor of the upward FAC in the center of the polar cap. However, there is no convincing data in the dusk-midnight sector that supports the circular plasma convection looking like “Hurricane”. The MHD simulation well reproduced the upward FAC in the center of the polar cap and downward FACs in the dawn sector. But, the reproduced circulation of ionospheric plasma extends equatorward in the dusk-midnight sector, so that the pattern looks like the conventional round cell. I would like to conclude that the data is not enough to verify the “Hurricane”.

Reply: We thank you for pointing out the need for further evidence of the auroral spot surrounded by fast flow. We have now overlapped the DMSP orbit with cross-track horizontal plasma flow vectors in each auroral image in a movie (see the examples shown in the following Figure R1). The movie indeed shows that the space hurricane is surrounded by circular fast flow at all local times of the DMSP satellite. Results from the 3D simulation (see the examples shown in the following Figure R2) and the AMPERE FACs maps further support that the space hurricane is surrounded by fast flows at all local times, especially supported by the clockwise rotations of the “arms” like FACs with circular plasma flows in the simulations (shown in an example at 14:00 UT in Figure R2).

Firstly, the plasma velocity shear around the center of the hurricane along one direction (e.g, DMSP orbit) is small, suggesting a weak FAC or weak acceleration. However, observations show peak FAC and acceleration around the center (this is different from the low atmosphere hurricane), suggesting the acceleration caused by plasma shears at different MLT is co-added at the center, and therefore, the convection is likely a complete circle.

Secondly, although the simulation only shows an upward FAC in the center of the polar cap and downward FACs in the dawn sector in the XY plane at $Z=8R_E$, the upward FAC spot surrounded by the downward FACs in the XY plane at $Z=4R_E$ (see

the examples shown in the following Figure R2), which may be due to the processes of keeping current continuity.

We added the following sentences:

“...The drift vectors (perpendicular to the spacecraft orbit) in Figure 2a (mauve) and Figure 2c show the cross-track horizontal (nearly north-south direction) ionospheric plasma drift from DMSP F16. ...Note that there will be a small horizontal offset between the in situ plasma drift data and the auroral images, because the converging magnetic field will cause the flow shears to decrease in horizontal extent from the DMSP in situ observation altitude (860 km) to the auroral mapping altitude (110km, Figure 2a). These flow shears give a clockwise circulation of ionospheric flow, which appears opposite to the rotation that might be inferred from the multiple “arms” of the auroral spot. This indicates an interesting difference from tropospheric hurricanes that is discussed latter....”.

Figure R1. Four selected DMSP SSUSI images of the space hurricane from four different satellites with overlapping the observed FAC and horizontal cross-track ion velocity along the satellite orbits.

Figure R2. Two selected snapshots of the simulated 2-D distribution of FACs and plasma velocity vectors in the X-Y plane at $Z=8 R_E$ and at $Z=4 R_E$, respectively. The format of each panel is same as Figure 3c.

[Minor comments]

Figure 4a.

The plasma convection is associated with the radial electric field converging to the upward FAC, which does not allow the radial flow of plasma.

Reply: Yes, there are transverse plasma flows as a reversed cell surrounding the upward FAC spot.

Figure 4b

Please make clear the relationship between the reconnection site and the generator that supplies the field-aligned currents. The reconnection may have created magnetic stresses on one side of the funnel, while there should be magnetic stresses on the other side of the funnel to complete the FACs.

Reply: We thank you for pointing out this for us. We now discussed the relationship between the reconnection site and the generator that supplies the FACs based on the suggestions from you and reviewer #2, and added the following sentences:

“During a northward IMF with a dominant B_y component, magnetic reconnection occurs between IMF and the Earth’s open magnetic field lines tailward of the cusp in the afternoon sector^{24, 3, 4, 12, 15} (high-latitude lobe reconnections, Figure 3b and S2, Movie S4). The newly reconnected open field lines are draped by the solar wind to move downward and then tailward from the morning side to the afternoon side in the high-latitude lobe region²⁴. During their downward and tailward motion, an elongated FAC sheet forms due to the flow shear, and the magnetosheath ions precipitate into the cusp ionosphere along field lines to give the downward FACs (like traces of dropping sands from a moving hourglass). In order to maintain current continuity in the ionosphere, the system sets up an upward FAC with a parallel potential that accelerates the existing electrons into the ionosphere and creates an “arm” of the auroral spot^{5, 15} observed by DMSP F16 in Figure 2a and shown in Figure 4a....”.

Line 90.

The “anti-clockwise” is not consistent with the clockwise flow in Figure 4a.

Reply: Yes, the arms of the auroral spot indeed show the trend of anti-clockwise rotation, which is because the aurora arms are formed by the particle precipitation and is represents the trace of reconnected magnetic field lines. This is confirmed by the clockwise motions of the reconnected magnetic field lines by a roughly steady lobe magnetic reconnections from our simulations. These motions result in a funnel formed for large and rapid deposition of solar wind/magnetosphere energy and momentum into the ionosphere around the magnetic pole unlike the expectation during extremely quiet geomagnetic conditions. We added the following sentences:

“...Inside the funnel, a corkscrew magnetic field forms with circular flow and upward FACs, which accelerate electrons that precipitate into the ionosphere^{15, 23, 30} and create the auroral spot with multiple “arms” as observed by DMSP F16 in Figure 2a. In other words, the auroral “arms” represent the trace of the footprints of the reconnected magnetic field lines, and shows an illusional trend of anti-clockwise rotation, which is opposite to the flow circulation and different from tropospheric hurricanes..”.

Lines 175-176

The field-aligned electric field is needed to accelerate electrons, but the FAC is generated by a dynamo or generator in the magnetosphere, which may be activated by the lobe reconnection. Since the FACs are well reproduced by the simulation code employed in the present paper, we could have an idea about the dynamo.

Reply: We thank you for pointing out this for us. Yes, the high-latitude lobe reconnection leads to the precipitation of magnetosheath ions into the cusp ionosphere along field lines to form downward FACs (shown in light green between the “arms” of the auroral spot in Figure 4a). In order to maintain current continuity in the ionosphere, the system sets up an upward FAC with a parallel potential that accelerates the existing electrons into the ionosphere and subsequently creates the multiple “arms” of the auroral spot. Thus, we suggested in the abstract that “the space hurricane is generated by a steady high-latitude lobe magnetic reconnection^{3,4} and the current continuity⁵ during a several hour period of northward interplanetary magnetic field (IMF) and very low solar wind density and speed...”.

We thank you again for your positive and constructive comments on our paper.

The end.

REVIEWER COMMENTS

Reviewer #1 (Remarks to the Author):

I am mostly satisfied by the authors' revisions in response to my comments. I would recommend this paper for publication in Nature Communications after the following minor changes:

Line 190: I suggest you use Figure 4 to show the reader how a reconnected field line "tends to move downward and then tailward from the morning side to the afternoon side in the high-latitude lobe region". It would be very nice to show the time evolution of a single field line that is executing this motion. In the current version of the video it is difficult to identify a field line moving as you describe it.

Figure S2: In my first round of comments I asked about the two different reconnection sites, which was also brought up by another reviewer. I suggest the authors can indicate with a star or other marking the approximate location of the reconnection site(s) in Figure S2.

Brandon Burkholder

Reviewer #2 (Remarks to the Author):

The authors addressed all my prior concerns. I'm especially happy with the quantitative comparison of the energy transport and comparison to other geomagnetic conditions. Overall I recommend publication after a few minor issues will be addressed by the authors.

Line 73: grammar

Line 75: What do you mean with "low solar wind conditions"?

Line 100: grammar

Line 222: What evidence do you have that reconnection is pulsed and not quasi-steady?

Table 1, column 8: The authors list average energies like 247.98 eV without giving error estimates. I highly doubt that you can determine average energies with an accuracy of 0.01 eV.

Harald Frey

Reviewer #4 (Remarks to the Author):

This paper reports a "hurricane-type" aurora, which belongs to the previously known type of aurora, high-latitude dayside auroral (HiLDA). I understand that the newly found features are as follows: (1) The "hurricane-type" aurora is embedded in the upward field-aligned current region. (2) The aurora is associated with the westward (clockwise when viewed above the north pole), rapid (exceeding 2 km/s) plasma convection. (3) The aurora has substructures, called "arms". They rotate eastward (counterclockwise), which is opposite to the ambient plasma convection. (4) The aurora is associated with large-amount of energy transfer from the solar wind to the ionosphere. This type of aurora is interesting, and as far as I know, this is the first manuscript that associate this type of aurora with energy transfer. Basically, the subject of this manuscript is suitable for possible publication in Nature Communications. However, I have 3 concerns about the term "space hurricane", the explanation of the "arms", and proper citing, and inappropriate citation.

[1] On the term "space hurricane":

The authors named the aurora as "space hurricane" because it carries large-amount of energy to the ionosphere, and the auroral shape is similar to hurricanes. The authors claim that it leads to rapid deposition of energy into the polar ionosphere. I agree that energy flux is extremely large in the heart of the aurora. I am wondering to know if the total amount of the energy is larger than that during the southward IMF condition. I recommend the authors to integrate the energy flux over the approximate size, and evaluate it with the typical values during the southward IMF condition. In general, during the southward IMF condition, the two-cell convection (which may also be regarded as "hurricanes") develops well, which carries large-amount of energy to the polar ionosphere, of the order of 10^{10} - 10^{11} W.

Regardless of the largeness of the energy deposit, I would recommend renaming the title by, like "A hurricane-like aurora...", and avoiding the unnecessary use of "space hurricane", since the term "space hurricane" will potentially give rise to misunderstanding because of the following reasons. First, the ionospheric flow velocity is not so extremely high. Secondly, the physical mechanism is totally different from tropospheric hurricanes (arising from low air pressure). Thirdly, it may challenge the common sense that the strongest energy deposition occurs during the southward IMF condition, and that the storms and substorms are the most significant disturbances in the magnetosphere and the ionosphere. The hurricane-like aurora is still interesting, and is worth publishing in Nature Communications.

[2] On the explanation of the "arms":

The authors explain the rotation of the "arms" in terms of the footprints of the reconnected magnetic field lines. If so, multiple reconnection, or intermittent reconnection should take place, and the reconnection point should rotate during the interval of the observation. I cannot see such features in the MHD simulation results shown in Supplement, as far as I understand. I am wondering to know if the counterclockwise motion of the "arms" (which is opposite to the ambient plasma motion) can be explained in terms of excess charge of the Hall current in the ionosphere. Ebihara and Tanaka (doi:10.1002/2015JA021697, doi:10.1088/1361-6587/aa89fd) have suggested that an auroral surge can travel in the direction of the Hall current (which is opposite to the ambient plasma motion) due to the excess charge (see Figure 7 of doi:10.1088/1361-6587/aa89fd). This may reasonably explain the counterclockwise motion of the "arms", since the Hall current flows counterclockwise (eastward) in the "arms" as shown in Figure 2. The important thing is that this process (excess charge) can occur in the dark ionosphere. I recommend verifying the "arms" is in darkness (at 100 km altitude). If this is the case, the "arms" forms by the requirement of the ionosphere.

[3] On inappropriate citation:

The authors should cite papers related to the energy transfer from the solar wind to the ionosphere during northward and southward IMF conditions. For the southward IMF condition, Fairfield and Cahill (doi:10.1029/JZ071i001p00155) pointed out the importance of the southward IMF. Perreault and Akasofu (doi:10.1111/j.1365-246X.1978.tb05494.x) suggested an empirical model for the energy coupling function. Vasyliunas et al. (doi:10.1016/0032-0633(82)90041-1) and Nishida (10.1007/BF00194626) pointed out that the solar wind kinetic energy can also be a source of the energy transferring to the magnetosphere. Tanaka (doi:10.1007/s11214-007-9168-4) showed the energy conversion near the magnetopause. Ebihara et al. (doi:10.1029/2018JA026177) demonstrated the energy transfer and conversion from the solar wind to the ionosphere.

Minor comments.

Line 37, 38, 39, 48, 49, 50:

"eV/(cm²·sr)" should be "eV/(cm²·s·sr)"? Please confirm the unit.

Line 61-62 "The stronger FACs occur near the hurricane center to maintain ionospheric current continuity in the presence of convergent ionospheric Pedersen currents...":

This statement sounds like that the Pedersen current is the cause, and the FAC is the result. From the observations, no one can say the cause and the effect of the FACs. In addition, FACs can be connected to the Hall current where the ionospheric conductivity is not uniform and/or for non-

steady condition. To avoid unnecessary confusion regarding the causality, it would be safe to say that "For the quasi-steady condition and the ionospheric conductivity is uniform, the large-scale, stronger FACs near the hurricane center can be connected to the convergent ionospheric Pedersen current ..."

Line 194-196 "The funnel of FAC appears as a spot with several "arms" and a trend of clockwise rotation":

This statement seems to be inconsistent with the description of the "arm" (Line 129-130).

Figure 4a may be misleading. Ion precipitation nor downward FACs are not clearly observed in the "arms" during the interval from 16:17 to 16:19 UT in Figure 2j.

Figure 4b may also be misleading. Do you have evidence that the magnetic field lines are so tightly twisted in the Earth's magnetosphere? Why the space hurricane is drawn as a cone, not being aligned with the magnetic field lines?

I took over original Reviewer #3's role. Reviewer #3 pointed out that the cartoon shown in Figure 4a is inconsistent with the observations, in particular, the flow pattern on the dusk side. However, the authors mentioned the flow shear near the center of the hurricane in their rebuttal letter. This is improper response to his/her concern. From left to right, clockwise, clockwise and counterclockwise vortices are drawn in Figure 4a. If the ionospheric conductivity is uniform, they will correspond to upward, upward, and downward field-aligned currents (FACs). Hereinafter, I assume that the dusk is to the left, and the dawn is to the right in Figure 4a. In Figure 4a, the left vortex (the dusk one, probably corresponding the upward FAC in the dusk-midnight sector) is so strong, and the strong flow shear appears in the region between the left and the center vortices. However, the left and center vortices (two clockwise vortices) should be canceled by each other in between (near the outer edge of the purple circle), which result in the longitudinally elongated convection (to the dusk-midnight sector) as shown by the solid contour in Figure 2b. This is what Reviewer #3 pointed out, I think. I recommend the authors to improve Figure 4a to resolve the inconsistency with the observation. I also recommend indicating the direction to the Sun in Figure 4a. Reviewer #3 introduced the terms "crescent cell" and "round cell" under northward IMF and B_y being not equal to zero. The cause of these two cells are not so simple as illustrated in Figure 4b. I strongly recommend citing Tanaka (1999, doi:10.1029/1999JA900077) who interpreted the formation of these cells, and improving Figure 4b to be more realistic, since the generation of the round cell is a key in the formation of the space hurricane. As for the minor comments, I believe that the authors respond to Reviewer #3's concerns fairly well.

Response to Reviewer #1:

We thank Dr. Brandon Burkholder again for reviewing our manuscript and providing positive and constructive comments. We have carefully considered your points for this revision. Below we provide point-by-point responses.

I am mostly satisfied by the authors' revisions in response to my comments. I would recommend this paper for publication in Nature Communications after the following minor changes:

Line 190: I suggest you use Figure 4 to show the reader how a reconnected field line "tends to move downward and then tailward from the morning side to the afternoon side in the high-latitude lobe region". It would be very nice to show the time evolution of a single field line that is executing this motion. In the current version of the video it is difficult to identify a field line moving as you describe it.

Reply: Thanks for the suggestion. Adding reconnected field lines in Figure 4 to show their evolution will make Figure 4b too busy. Instead, we added a arrowed curve and numbered the reconnected field lines in Figure 3b to illustrate the evolution and created a higher time resolution video (Movie S4), which clearly shows the evolution of the reconnected field lines.

Figure 3b 3-D distribution of selected magnetic field lines with magenta crosses representing the reconnection sites.

Figure S2: In my first round of comments I asked about the two different reconnection sites, which was also brought up by another reviewer. I suggest the authors can indicate with a star or other marking the approximate location of the reconnection site(s) in Figure S2.

Reply: We thank you for this suggestion. We have added the reconnection sites as a magenta cross in both Figure S2 and Figure 3b.

Figure S2. Two different views of the 3-D selected magnetic field lines simulated by the PPMLR-MHD code for the snapshot shown in Figure 3. The format is the same as Figure 3b. The magenta crosses represent the reconnection sites.

We thank you again for your constructive comments that helped to improve the manuscript.

Response to Reviewer #2:

We thank Dr. Harald Frey for reviewing our manuscript and providing positive and further constructive comments. We have carefully considered your points for this revision. Below we provide point-by-point responses.

The authors addressed all my prior concerns. I'm especially happy with the quantitative comparison of the energy transport and comparison to other geomagnetic conditions. Overall I recommend publication after a few minor issues will be addressed by the authors.

Reply: We thank you for the suggestion on quantitative comparison of the energy transportation, which indicates the importance of the processes associated with the "Space Hurricanes"

Line 73: grammar

Reply: We have improved the sentence as: "On 20th August 2014, a relatively stable northward IMF condition (IMF Bz > 0 for more..."

Line 75: What do you mean with "low solar wind conditions"?

Reply: It refers to "low solar wind speed and density". The term "low solar wind conditions" has been replaced by "...roughly stable interplanetary conditions with low solar wind speed and density...".

Line 100: grammar

Reply: We have improved the sentence as: "... S1 and S2 and Figure 1), same as the disappearance of the HiLDA spots¹¹."

Line 222: What evidence do you have that reconnection is pulsed and not quasi-steady?

Reply: We don't have strong evidence to support that reconnection is pulsed and not quasi-steady. Therefore, we used "pulsed or quasi-steady" instead.

Table 1, column 8: The authors list average energies like 247.98 eV without giving error estimates. I highly doubt that you can determine average energies with an accuracy of 0.01 eV.

Reply: The values are all come from the average processes and not related to the accuracy of the measurements. We now more appropriately rounded the values as "248", "709", "988", etc.

We thank you again for your valuable and constructive comments.

Response to Reviewer #4:

We thank the reviewer for many positive and constructive comments. We have carefully revised our manuscript based on the comments. Please find below our replies to each of your comments.

This paper reports a "hurricane-type" aurora, which belongs to the previously known type of aurora, high-latitude dayside auroral (HilDA). I understand that the newly found features are as follows: (1) The "hurricane-type" aurora is embedded in the upward field-aligned current region. (2) The aurora is associated with the westward (clockwise when viewed above the north pole), rapid (exceeding 2 km/s) plasma convection. (3) The aurora has substructures, called "arms". They rotate eastward (counterclockwise), which is opposite to the ambient plasma convection. (4) The aurora is associated with large-amount of energy transfer from the solar wind to the ionosphere. This type of aurora is interesting, and as far as I know, this is the first manuscript that associate this type of aurora with energy transfer. Basically, the subject of this manuscript is suitable for possible publication in Nature Communications.

Reply: We thank the reviewer for his/her appreciation on the potential contribution of our manuscript.

However, I have 3 concerns about the term "space hurricane", the explanation of the "arms", and proper citing, and inappropriate citation.

[1] On the term "space hurricane":

The authors named the aurora as "space hurricane" because it carries large-amount of energy to the ionosphere, and the auroral shape is similar to hurricanes. The authors claim that it leads to rapid deposition of energy into the polar ionosphere. I agree that energy flux is extremely large in the heart of the aurora. I am wondering to know if the total amount of the energy is larger than that during the southward IMF condition. I recommend the authors to integrate the energy flux over the approximate size, and evaluate it with the typical values during the southward IMF condition. In general, during the southward IMF condition, the two-cell convection (which may also be regarded as "hurricanes") develops well, which carries large-amount of energy to the polar ionosphere, of the order of 10^{10} - 10^{11} W.

Reply: We thank the reviewer for this suggestion. We have calculated a DMSP polar pass of a typical southward IMF cases during non-storm time, which has about 4.6 times smaller time integrated JE (Σ JE) than that for the pass with the space hurricane and also has about 8.3 times smaller average energy flux (see table 1). The IMF is strongly southward during the super storm case, and this case has only about 4.6 times higher time integrated JE (Σ JE) than for the pass with space hurricane, but has about 3.2 times smaller average energy flux. Thus, we believe it is fair to say that the space hurricane causes extremely large deposition of energy into the polar ionosphere.

Regardless of the largeness of the energy deposit, I would recommend renaming the

title by, like "A hurricane-like aurora...", and avoiding the unnecessary use of "space hurricane", since the term "space hurricane" will potentially give rise to misunderstanding because of the following reasons. First, the ionospheric flow velocity is not so extremely high. Secondly, the physical mechanism is totally different from tropospheric hurricanes (arising from low air pressure). Thirdly, it may challenge the common sense that the strongest energy deposition occurs during the southward IMF condition, and that the storms and substorms are the most significant disturbances in the magnetosphere and the ionosphere. The hurricane-like aurora is still interesting, and is worth publishing in Nature Communications.

Reply: We thank the reviewer for this interesting suggestion, and we have thought about it carefully. We would like to summarize the main characters of the structure we called "space hurricane" here first:

The space hurricane is associated with 1) a huge "cyclone-like" auroral spot (diameter over 1000 km) with multiple "arms", 2) a spot-like strong upward FAC, 3) zero horizontal flow near its center (the hurricane eye) as well as strong flow shears around the edges, 4) ion upflows, 5) enhanced electron temperature (about 1000 K enhancement), 6) a negative-to-positive bipolar magnetic structure, (implying a circular magnetic field perturbation), 7) strong energy deposition and 8) electron inverted-V acceleration. Thus, the space hurricane has not only its auroral feature but also plasma and magnetic structures, which are beyond what could be properly and fully represented by "a hurricane-like aurora". Below we provided responses to your three reasons:

Firstly, although the ionospheric flow velocities around the space hurricane are not extremely high, the velocities are still very high (up to about 2100 km/s in sunward at its duskside and 800 km/s in anti-sunward at its dawnside) and circular under extremely quiet interplanetary conditions.

Secondly, we agree that the physical mechanism is totally different from tropospheric hurricanes (arising from low air pressure). Here we just named it in analogy based on its main characters, which is learnt from the terminology of "storm" and "substorm", and "space weather", etc.

Thirdly, the space hurricane causes strong energy deposition, even stronger than a typical southward IMF condition during a non-storm time, though not as strong as a major storm or strong substorm, because of scale size (and duration for a storm). This is in analogy with a hurricane, which does not have the scale size or large-geographic impact of a major mid-latitude cyclone.

Based on the above considerations, we have decided to stay with the term 'space hurricane'. As a by-product, maybe this title would be helpful for public appreciation of the space weather phenomena.

[2] On the explanation of the "arms":

The authors explain the rotation of the "arms" in terms of the footprints of the reconnected magnetic field lines. If so, multiple reconnection, or intermittent reconnection should take place, and the reconnection point should rotate during the

interval of the observation. I cannot see such features in the MHD simulation results shown in Supplement, as far as I understand. I am wondering to know if the counterclockwise motion of the "arms" (which is opposite to the ambient plasma motion) can be explained in terms of excess charge of the Hall current in the ionosphere. Ebihara and Tanaka (doi:10.1002/2015JA021697, doi:10.1088/1361-6587/aa89fd) have suggested that an auroral surge can travel in the direction of the Hall current (which is opposite to the ambient plasma motion) due to the excess charge (see Figure 7 of doi:10.1088/1361-6587/aa89fd). This may reasonably explain the counterclockwise motion of the "arms", since the Hall current flows counterclockwise (eastward) in the "arms" as shown in Figure 2. The important thing is that this process (excess charge) can occur in the dark ionosphere. I recommend verifying the "arms" is in darkness (at 100 km altitude). If this is the case, the "arms" forms by the requirement of the ionosphere.

Reply: We thank the reviewer for his/her nice suggestion. Unfortunately, our case is not in darkness, because the space hurricane happened on 20 August 2014 which is under sunlight conditions and far poleward from the terminator at 100 km altitude. Thus, we think the excess charge mechanism may not work for our case because the space hurricane may not cause the strong gradient in ionospheric conductivity required for the process of excess charge. For the explanation of rotation of the "arms" in terms of the footprints of the reconnected magnetic field lines, we only find that a pulsed or quasi-steady reconnection is taking place in the northern lobe regions from the MHD simulation (see the updated Movie S4), which may be because the IMF and solar wind condition is roughly stable. It is the magnetic tension that causes the rotation of reconnected field lines, rather than the rotation of the reconnection point. Thus, we would like to still explain the rotation of the "arms" in terms of the footprints of the magnetic field lines following their reconnections.

[3] On inappropriate citation:

The authors should cite papers related to the energy transfer from the solar wind to the ionosphere during northward and southward IMF conditions. For the southward IMF condition, Fairfield and Cahill (doi:10.1029/JZ071i001p00155) pointed out the importance of the southward IMF. Perreault and Akasofu (doi:10.1111/j.1365-246X.1978.tb05494.x) suggested an empirical model for the energy coupling function. Vasyliunas et al. (doi:10.1016/0032-0633(82)90041-1) and Nishida (10.1007/BF00194626) pointed out that the solar wind kinetic energy can also be a source of the energy transferring to the magnetosphere. Tanaka (doi:10.1007/s11214-007-9168-4) showed the energy conversion near the magnetopause. Ebihara et al. (doi:10.1029/2018JA026177) demonstrated the energy transfer and conversion from the solar wind to the ionosphere.

Reply: We thank the reviewer for his/her nice suggestion. We have updated our citations by including all of the references suggested by the reviewer.

Minor comments.

Line 37, 38, 39, 48, 49, 50:

"eV/(cm²·sr)" should be "eV/(cm²·s·sr)"? Please confirm the unit.

Reply: Yes, the unit is "eV/(cm²·sr)" because it is time integrated total energy flux, which is total energy flux JE (eV/(cm²·s·sr)) multiplied by time.

Line 61-62 "The stronger FACs occur near the hurricane center to maintain ionospheric current continuity in the presence of convergent ionospheric Pedersen currents...":

This statement sounds like that the Pedersen current is the cause, and the FAC is the result. From the observations, no one can say the cause and the effect of the FACs. In addition, FACs can be connected to the Hall current where the ionospheric conductivity is not uniform and/or for non-steady condition. To avoid unnecessary confusion regarding the causality, it would be safe to say that "For the quasi-steady condition and the ionospheric conductivity is uniform, the large-scale, stronger FACs near the hurricane center can be connected to the convergent ionospheric Pedersen current ..."

Reply: We thank the reviewer for his/her nice suggestion. We have followed his/her suggestion to replace the sentence of "The stronger FACs occur near the hurricane center to maintain ionospheric current continuity in the presence of convergent ionospheric Pedersen currents..." by "For the quasi-steady condition and the uniform ionospheric conductivity, the large-scale, stronger FACs near the hurricane center can be connected to the convergent ionospheric Pedersen current ...".

Line 194-196 "The funnel of FAC appears as a spot with several "arms" and a trend of clockwise rotation":

This statement seems to be inconsistent with the description of the "arm" (Line 129-130).

Reply: We thank the reviewer for noticing this. We have revised the sentence as "The funnel of FAC appears as a spot with several "arms" and a trend of anticlockwise rotation"

Figure 4a may be misleading. Ion precipitation nor downward FACs are not clearly observed in the "arms" during the interval from 16:17 to 16:19 UT in Figure 2j.

Reply: We thank the reviewer for pointing out this for us. We have removed the ion precipitation from Figure 4a.

Figure 4b may also be misleading. Do you have evidence that the magnetic field lines are so tightly twisted in the Earth's magnetosphere? Why the space hurricane is drawn as a cone, not being aligned with the magnetic field lines?

Reply: We thank the reviewer for pointing out this for us. Firstly, we have plotted out the magnetic field measurement (subtracting the modeled background) from the satellite which shows a negative-to-positive bipolar magnetic structure (varying about 1000nT), and implies a circular magnetic field perturbation. Thus we draw it as a distorted cone. Secondly, we improved the cone to make it along the field lines and

the field lines inside the cone to make them not too twisted in the new Figure 4b.

Figure 2f the three components of the measured magnetic field subtracted by the modeled magnetic field from the International Geomagnetic Reference Field (IGRF) model³³.

Figure 4b Schematic of the 3-D magnetosphere when a space hurricane happened.

I took over original Reviewer #3's role. Reviewer #3 pointed out that the cartoon shown in Figure 4a is inconsistent with the observations, in particular, the flow pattern on the dusk side. However, the authors mentioned the flow shear near the center of the hurricane in their rebuttal letter. This is improper response to his/her concern. From left to right, clockwise, clockwise and counterclockwise vortices are drawn in Figure 4a. If the ionospheric conductivity is uniform, they will correspond to upward, upward, and downward field-aligned currents (FACs). Hereinafter, I assume that the dusk is to the left, and the dawn is to the right in Figure 4a. In Figure 4a, the left vortex (the dusk one, probably corresponding the upward FAC in the dusk-midnight sector) is so strong, and the strong flow shear appears in the region between the left and the center vortices. However, the left and center vortices (two clockwise vortices) should be canceled by each other in between (near the outer edge of the purple circle), which result in the longitudinally elongated convection (to the dusk-midnight sector) as shown by the solid contour in Figure 2b. This is what Reviewer #3 pointed out, I think. I recommend the authors to improve Figure 4a to

resolve the inconsistency with the observation. I also recommend indicating the direction to the Sun in Figure 4a. Reviewer #3 introduced the terms "crescent cell" and "round cell" under northward IMF and B_y being not equal to zero. The cause of these two cells are not so simple as illustrated in Figure 4b. I strongly recommend citing Tanaka (1999, doi:10.1029/1999JA900077) who interpreted the formation of these cells, and improving Figure 4b to be more realistic, since the generation of the round cell is a key in the formation of the space hurricane. As for the minor comments, I believe that the authors respond to Reviewer #3's concerns fairly well.

Reply: We thank the reviewer for pointing out this for us. We have carefully read the reference and improved the Figure 4a by following the reviewer's suggestions. The new figure 4a is attached below:

Figure 4a Schematic of a space hurricane in the northern polar ionosphere.

We thank you again for your positive and constructive comments on our paper.

REVIEWERS' COMMENTS

Reviewer #1 (Remarks to the Author):

I am satisfied with the authors revisions.

Reviewer #2 (Remarks to the Author):

The authors addressed all my prior concerns about their first revised version and I'm happy with the result. I recommend publication as is.

Reviewer #4 (Remarks to the Author):

I confirmed that the authors responded my concerns almost properly. I have only a minor comment regarding the amount of the electron precipitation. In Line 154-159, it is stated that the averaged energy flux (2.2×10^{12} eV/cm² s sr) is much larger than during typical quiet and super storms in the same region. Readers may consider (or misunderstand) that the space hurricane is accompanied with extreme electron precipitation, which is much intense in comparison with substorms and storms. (I understand that the authors compared the electron energy flux in the same region). It would be safe (and fair) to add the statement regarding the electron precipitation at lower latitudes. The following is an example that is supposed to be added at the end of the paragraph (Line 157-159):

"A large number of electrons are known to continuously precipitate into the auroral oval during substorms and storms. The electron flux (2.2×10^{12} eV/(cm² s sr)) in the space hurricane is much higher than during substorm expansion (Wing et al., 2013, 10.1002/jgra.50160), but is comparable to that during super storms (Shiokawa et al., 10.1029/96GL01955). During extremely super storms, the flux can exceed 10^{13} eV/(cm² s sr) (Shiokawa et al., 10.1029/96GL01955; Ebihara et al., 10.1002/2017SW001693). These observations were made in the auroral oval (namely at much lower latitudes), and are quite different from the space hurricane."

Except for this comment, I am fully satisfied the revision. I admire the authors' tenacious effort.

Response to Reviewer #4:

We thank the reviewer again for reviewing our manuscript and providing constructive suggestions. We have carefully considered your points for this revision. Below we provide point-by-point responses.

I confirmed that the authors responded my concerns almost properly. I have only a minor comment regarding the amount of the electron precipitation. In Line 154-159, it is stated that the averaged energy flux (2.2×10^{12} eV/cm² s sr) is much larger than during typical quiet and super storms in the same region. Readers may consider (or misunderstand) that the space hurricane is accompanied with extreme electron precipitation, which is much intense in comparison with substorms and storms. (I understand that the authors compared the electron energy flux in the same region). It would be safe (and fair) to add the statement regarding the electron precipitation at lower latitudes. The following is an example that is supposed to be added at the end of the paragraph (Line 157-159):

"A large number of electrons are known to continuously precipitate into the auroral oval during substorms and storms. The electron flux (2.2×10^{12} eV/(cm² s sr)) in the space hurricane is much higher than during substorm expansion (Wing et al., 2013, 10.1002/jgra.50160), but is comparable to that during super storms (Shiokawa et al., 2010, 10.1029/96GL01955). During extremely super storms, the flux can exceed 10^{13} eV/(cm² s sr) (Shiokawa et al., 2010, 10.1029/96GL01955; Ebihara et al., 2010, 10.1002/2017SW001693). These observations were made in the auroral oval (namely at much lower latitudes), and are quite different from the space hurricane."

Except for this comment, I am fully satisfied the revision. I admire the authors' tenacious effort.

Reply: We thank the reviewer for this nice suggestion. We have added the sentences in the main text by following his/her suggestions as follows:

"...The space hurricane has an average energy flux about 5.5 times higher than its own whole polar pass, and this whole pass is about 15.1 times higher than the pass for the typical quiet case, about 8.3 times higher than the pass for the typical southward IMF case, and even 3.2 times higher than the super storm case (see Table 1). These means that the average electron energy flux in the space hurricane (2.2×10^{12} eV/(cm²·s·sr)) is much higher than that during substorm expansion³³, but is comparable to that during super storms (sometimes exceeding 10^{13} eV/(cm²·s·sr) during strikingly super storms)^{34,35}. Note that the large electron precipitation flux during substorms and storms is within the auroral oval, which is located at much lower latitudes than the space hurricane....".

We thank you again for your constructive comments that helped to improve the manuscript.